

# Clock factorized quantum Monte Carlo method for long-range interacting systems

Zhijie Fan[1,2★], Chao Zhang[1,2†] and Youjin Deng[1,2,3‡]

**1** Department of Modern Physics, University of Science and Technology of China,
Hefei, Anhui 230026, China
**2** Hefei National Laboratory, University of Science and Technology of China,
Hefei 230088, China
**3** MinJiang Collaborative Center for Theoretical Physics,
College of Physics and Electronic Information Engineering,
Minjiang University, Fuzhou 350108, China

★ zfanac@ustc.edu.cn , † zhangchao1986sdu@gmail.com , ‡ yjdeng@ustc.edu.cn

## Abstract

Simulating long-range interacting systems is a challenging task due to its computational complexity that the computational effort for each local update is of order $\mathcal{O}(N)$, where $N$ is the size of the system. In this work, we introduce the clock factorized quantum Monte Carlo method, an efficient technique for simulating long-range interacting quantum systems. The method is developed by generalizing the clock Monte Carlo method for classical systems [Phys. Rev. E 99 010105 (2019)] to the path-integral representation of long-range interacting quantum systems, with some specific treatments for quantum cases and a few significant technical improvements in general. We first explain how the clock factorized quantum Monte Carlo method is implemented to reduce the computational overhead from $\mathcal{O}(N)$ to $\mathcal{O}(1)$. In particular, the core ingredients, including the concepts of bound probabilities and bound rejection events, the recursive sampling procedure, and the fast algorithms for sampling an extensive set of discrete and small probabilities, are elaborated. Next, we show how the clock factorized quantum Monte Carlo method can be flexibly implemented in various update strategies, like the Metropolis and worm-type algorithms. Finally, we demonstrate the high efficiency of the clock factorized quantum Monte Carlo algorithms using examples of three typical long-range interacting quantum systems, including the transverse field Ising model with long-range $z$-$z$ interaction, the extended Bose-Hubbard model with long-range density-density interactions, and the XXZ Heisenberg model with long-range spin interactions. We expect that the clock factorized quantum Monte Carlo method would find broad applications in statistical and condensed-matter physics.

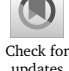

# 1   Introduction

Markov-chain Monte Carlo methods (MCMC) are highly valuable tools across numerous fields of science and engineering [1–8], particularly for estimating high-dimensional integrals. These methods rely on statistical sampling approaches that generate a large number of random configurations of the system being studied. Each configuration has a stationary distribution or weight, which is usually a Boltzmann distribution. The generation of subsequent configurations depends on the resulting changes in energy. These configurations are then used to estimate the properties of the system, such as its energy and other observables.

Despite a long history, the founding Metropolis algorithm remains the most successful and influential MCMC method due to its generality and ease of use. It is a family of MCMC methods that adopt local update strategies and the so-called Metropolis acceptance filter. Quantum Monte Carlo (QMC) methods using local updating schemes are a powerful tool for studying quantum systems and have continued to evolve with the development of numerous algorithms, such as path-integral Monte Carlo (PIMC) [9,10], stochastic series expansion (SSE) [11], variational Monte Carlo (VMC) [12–14], diffusion Monte Carlo (Diffusion MC) [15], determinant Monte Carlo (detMC) [16,17], Diagrammatic Monte Carlo (DiagMC) [18–20] and so on. QMC has been successfully applied to various systems, including the Hubbard model [21,22], the $t-J$ model [14], the polaron model [19], Ising, $XY$, and the Heisenberg model [23,24].

Despite the significant advancements made, there remain several challenges that are yet to be overcome in computational simulations. The core challenging problem in computational simulations is so-called the exponential wall. One example of this problem in classical systems is to simulate spin-glass systems, where the free energy landscape of the systems is characterized by a large number of local minima, or energy valleys, separated by high energy barriers, leading to exponentially increasing computational cost as the system size increases. As the system is cooled to lower temperatures, it becomes increasingly difficult to escape from these local minima and find the true ground state. In the quantum case, a similar problem is the sign problem, which arises when QMC algorithms have to generate negative weights for certain configurations, leading to inaccurate estimates of the expectation value of observables.

The second challenge lies in simulations experiencing critical slowing-down as they approach phase transitions, where nearby samples can be highly correlated, and simulation efficiency decreases rapidly as the system size increases. Enormous effort has been devoted to circumventing this limitation. Various efficient update strategies have been designed, including the cluster [25, 26], direct-loop [27], event-chain [28], and worm algorithms [29].

Another challenge is the computational complexity associated with simulating systems with long-range interactions, which can require calculating the induced total energy change for each attempted move and lead to expensive computational costs of up to $\mathcal{O}(N)$ per local attempt, where $N$ is the system size. Several techniques are also available to reduce the computational complexity of specific algorithms and systems. In the worm algorithm with DiagMC [9], the attractive part of the pairwise potential energy is expanded into diagrammatic contributions, which affords a complete microscopic account of the long-range part of the potential energy while keeping the computational complexity of all updates independent of the size of the simulated system. In the cluster-updates scheme [30], an efficient sampling procedure is to place occupied bonds rather than visiting each bond sequentially and throwing a random number to decide its status. The event-chain Monte Carlo method combines the factorized Metropolis filter and Walker's alias and has primarily been successfully utilized in the fields of physics and chemistry [28, 31–33].

Recently, Ref [34] proposed a generic clock Monte Carlo method for classical systems, using the factorized Metropolis filter to reduce the computational complexity to $\mathcal{O}(1)$ and offers significant benefits in terms of simulation efficiency. The basis of the clock Monte Carlo method is the so-called factorized Metropolis filter proposed in Ref [35]. Unlike the Metropolis filter where the acceptance probability $P_{\text{Met}}$ is determined by the total induced energy change, the factorized Metropolis filter factorizes the acceptance probability as $P_{\text{fac}} = \prod P_j$, where factor $P_j$ is given by the induced energy change for the associated interaction term $j$. Namely, all interaction terms are treated independently, and each of them contributes a factor to the overall acceptance probability $P_{\text{fac}}$. As a consequence, in the stochastic determination of the fate (acceptance or rejection) of the attempted move, any single rejection from one of the factors, $P_j$, would be sufficient to reject the attempted move. Making use of the independence of these factors, one can define a set of first-rejection events and design a random process for sampling these first-rejection events. However, direct sampling of these events is computationally expensive because $P_j$ depends on the local configuration associated with the interaction term $j$. This obstacle is addressed by the clock technique which samples a set of bound first-rejection events independent of configurations and utilizes a resampling procedure to recover the original probability distribution for first-rejection events [34]. Note that there exist efficient algorithms for sampling configuration-independent discrete probability distributions with $\mathcal{O}(1)$ or $\mathcal{O}(\log N)$ computational efficiency, e.g., Walker's alias method or the thinning method. Thus, unlike the standard Metropolis filter of $\mathcal{O}(N)$ computational complexity, the factorized filter combined with the clock technique may lead to a sampling process of dramatically reduced effort. In short, thanks to the factorized Metropolis filter, the fate of an attempted move can be efficiently determined by a sampling process of first-rejection events, leading to a significant increase in simulation efficiency. The clock Monte Carlo method demonstrates $\mathcal{O}(1)$ computational complexity on several classical long-range interacting systems [34].

In this work, inspired by the clock Monte Carlo method for classical systems, we adopt the factorized Metropolis filter to the PIMC method and propose a generic Monte Carlo scheme for simulating long-range interacting quantum systems, which we call the clock factorized quantum Monte Carlo method.

First, we introduce the concept of the recursive clock sampling scheme, which can be considered as a generalization of the aforementioned clock technique. It can be interpreted as a recursive sampling process on a tree structure. We further show that the factorized Metropolis

filter and the recursive clock sampling technique can be properly applied to generic configuration weights and non-symmetric proposal probability. Moreover, implementing the recursive clock sampling scheme using the dynamic thinning method [36] is discussed in detail. We also note that, with the dynamic thinning method, the Luijten-Blöte cluster method can be significantly improved [37], of which the formulation becomes very simple and generic. In addition, there is no need to build a lookup table or use discrete cumulative probability integration approximations to sample bond generation events.

Second, we apply the recursive clock sampling method to the path-integral representation of quantum systems. Note that our method allows for the factorization of the non-diagonal term and the proposal probability associated with the update. Hence, the recursive clock sampling process can be integrated with various update strategies, including conventional Metropolis-type local updates, cluster updates, worm-type updates, etc, and it can deal with long-range interactions (diagonal terms) as well as long-range hopping amplitudes (non-diagonal terms). For the diagonal term, the dynamic thinning method can be applied when utilizing recursive clock sampling for the long-range interaction terms. For the non-diagonal term, when the dynamic thinning method is not directly applicable, we can combine Walker's alias method to increase the overall acceptance rate. Particularly, we consider three typical systems and apply the recursive clock sampling process in various update schemes: (i) the transverse field Ising model with long-range $z$-$z$ interactions using local Metropolis-type update, (ii) the extended Bose-Hubbard model with long-range density-density interaction using worm update, and (iii) the long-range XXZ Heisenberg model using worm update with long-range hopping. We perform extensive benchmark simulations on systems of various sizes $L$ in both two dimensions (2D) and three dimensions (3D) and achieve the expected $\mathcal{O}(1)$ computational efficiency. In particular, we demonstrate the overall efficiency improvement from $\mathcal{O}(N)$ to $\mathcal{O}(1)$, which takes into account the enhancement of computational complexity and the decrease of acceptance probability.

Finally, we mention that, in comparison with the standard Metropolis filter, the factorized Metropolis filter has a smaller acceptance probability since the energy compensation between different interaction terms is absent in the latter. This price is probably why the latter was proposed about 60 years later than the former. For a system that satisfies the absolute energy extensively, both the acceptance probabilities, $P_{\text{Met}}$ and $P_{\text{fac}}$, are of $\mathcal{O}(1)$, thus the price is minor [34]. However, for some frustrated systems with slowly-decaying interactions, the factorized probability $P_{\text{fac}}$ may decrease as system size increases. To (partially) overcome this problem, one can group several interactions that are likely to have energy compensation into a single factor such that their total induced energy change would benefit from energy compensation and lead to a higher acceptance probability. This trick is called the box technique [34]. The standard Metropolis filter is recovered in the limiting case that all the interaction terms are in a single box.

The clock factorized QMC method is expected to have wide-ranging applications in the field of physics with long-range interactions. For example, the Coulomb interaction between charged particles is a long-range interaction that plays a fundamental role in electrostatics. This interaction is responsible for many phenomena in physics, including the behavior of plasma and the formation of crystals [38,39]. Another essential interaction is the magnetic or electronic dipolar interaction, which plays an important role in the behavior of ferromagnetic materials [40–42]. In addition to these examples, long-range interactions can also have important effects on fluid dynamics. For instance, the van der Waals force between molecules is a long-range interaction that can cause fluids to condense into a liquid or solid phase [43]. The long-range Ising model with trapped-ion quantum simulators is another type of long-range interaction, which has the potential to advance our understanding of fundamental physics and to pave the way for new technologies such as quantum computing [44,45]. Understanding these

interactions is essential for comprehending many physical phenomena and developing new technologies. Our algorithm can be applied to various physical systems that involve long-range interactions, enabling researchers to obtain accurate and reliable results within a reasonable computational time in their simulations.

The rest of this paper is organized as follows. In Section 2, we introduce the basic idea of the recursive clock sampling technique, a generalized version of clock Monte Carlo technique [34]. In Section 3, we present the implementation of a recursive clock sampling scheme. Section 4 contains the clock factorized quantum Monte Carlo (clock factorized QMC) algorithms. Section 5 discusses more possible implementations of the clock factorized QMC method and concludes the paper.

## 2 Clock sampling for proposed updates

In this section, we first elaborate on the basic ingredients of the Clock Monte Carlo method [34] using a general configuration weight and introduce the recursive clock sampling framework. A key advancement is our integration of the factorized Metropolis filter into the Metropolis-Hastings criterion, which enables updates with asymmetric *a priori* probabilities and finite local fields. These improvements are crucial for incorporating off-diagonal terms and local diagonal terms in PIMC and thereby extend the method's applicability to a broader range of quantum systems.

### 2.1 Metropolis filter and computational complexity

Markov Chain Monte Carlo (MCMC) methods are powerful computational tools for simulating complex systems in diverse scientific fields [1–8]. They can efficiently sample complex, high-dimensional probability distributions that are difficult to generate directly. In physical simulations, MCMC generates a chain of configurations whose equilibrium distribution approximates the thermodynamic ensemble of the physical model. New configurations are generated via a *Markov process* in which the transition probability of the next configuration depends only on the preceding one. In order for MCMC to reach equilibrium, two conditions must be met: ergodicity and the global balance condition. Ergodicity demands that MCMC can eventually explore all possible configurations of the system, while the global balance condition requires the total flow into a configuration to equal the total flow out of it,

$$\sum_{\mathcal{S}'} \pi(\mathcal{S}) \mathcal{P}\left(\mathcal{S} \to \mathcal{S}'\right) = \sum_{\mathcal{S}'} \pi\left(\mathcal{S}'\right) \mathcal{P}\left(\mathcal{S}' \to \mathcal{S}\right), \tag{1}$$

where $\pi(\mathcal{S})$ ($\pi\left(\mathcal{S}'\right)$) is the probability weight of configuration $\mathcal{S}$ ($\mathcal{S}'$), and $\mathcal{P}\left(\mathcal{S} \to \mathcal{S}'\right)$ represents the transition probability from configuration $\mathcal{S}$ to $\mathcal{S}'$. In practice, instead of Eq. (1), the *detailed balance* condition is much more often imposed, which requires the flows between any two configurations to be equal,

$$\pi(\mathcal{S}) \mathcal{P}\left(\mathcal{S} \to \mathcal{S}'\right) = \pi\left(\mathcal{S}'\right) \mathcal{P}\left(\mathcal{S}' \to \mathcal{S}\right). \tag{2}$$

It is stronger than the global balance condition since it guarantees that the transitions between states are reversible, ensuring proper convergence to the target distribution.

*The Metropolis algorithm.* Among various MCMC methods, the Metropolis algorithm is probably the most successful and influential one. First introduced by Metropolis et al. in 1953 [46], this algorithm has significantly impacted numerous fields, including physics [1], computational chemistry [47], and Bayesian inference [48]. In the Metropolis algorithm, each elemental Markov step is executed in two sub-steps: proposal of a local update and stochastic

determination of the *fate* (acceptance or rejection) of the proposed update. In a transition from configuration $\mathcal{S}$, the algorithm proposes a new state $\mathcal{S}'$ and then decides whether to accept or reject the update based on an acceptance probability. The proposal sub-step exhibits both locality and symmetry. The locality implies that the new configuration $S'$ is selected from a finite range of configurations in the proximity of the initial configuration $\mathcal{S}$. Meanwhile, symmetry means that the likelihood of choosing $\mathcal{S}'$ from $\mathcal{S}$ is identical to that of $\mathcal{S}$ from $\mathcal{S}'$. Consider a physical system whose configurations obey Boltzmann distribution $\pi(\mathcal{S}) = \exp(-\beta E)$, where $\beta$ denotes the inverse temperature and $E$ is the total energy of the configuration. The acceptance probability for an update from $\mathcal{S}$ to $\mathcal{S}'$ is

$$P_{\text{Met}} = \min\left(1, \frac{\pi(\mathcal{S}')}{\pi(\mathcal{S})}\right) = \exp\left(-\beta\left[\Delta E_{\text{tot}}\right]^+\right), \tag{3}$$

with $[x]^+ \equiv \max(0, x)$ and $\Delta E = E(\mathcal{S}') - E(\mathcal{S})$ being the total energy difference between the two configurations. This expression, known as *the Metropolis filter*, satisfies the detailed balance condition in Eq. (2). In practice, the proposed update is accepted if a uniform random number $\texttt{ran} \in [0, 1)$ satisfies $\texttt{ran} < \text{P}_{\text{Met}}$. Otherwise, it is rejected.

The Metropolis-Hastings algorithm is a generalized Metropolis algorithm by introducing an *a priori* proposal distribution $\mathcal{A}(\mathcal{S} \to \mathcal{S}')$ [49]. The new configuration $\mathcal{S}'$ is proposed from $\mathcal{S}$ according to $\mathcal{A}(\mathcal{S} \to \mathcal{S}')$ and the transition probability becomes $\mathcal{P}(\mathcal{S} \to \mathcal{S}') = \mathcal{A}(\mathcal{S} \to \mathcal{S}') \times P\left(\mathcal{S} \to \mathcal{S}'\right)$. The acceptance probability is given by,

$$P_{\text{M-H}} = \min\left(1, \frac{\mathcal{A}(\mathcal{S}' \to \mathcal{S})}{\mathcal{A}(\mathcal{S} \to \mathcal{S}')} \frac{\pi(\mathcal{S}')}{\pi(\mathcal{S})}\right). \tag{4}$$

This algorithm allows more flexibility in proposal distribution, making it more efficient when sampling complex systems. In some cases, minor modifications in the algorithm, arising from a proper choice of $\mathcal{A}$, may lead to $\mathcal{O}(1)$ but significant improvement of efficiency.

*Computational complexity.* Despite its success in various domains, the Metropolis algorithm encounters a significant computational bottleneck when dealing with long-range interactions. Consider a long-range interacting classical system with $N$ sites, where each site interacts with the remaining $N-1$ sites, resulting in a total of $N(N-1)/2$ interacting pairs. At each step of the Metropolis algorithm, one randomly selects a site $i$ and updates its state. The induced total energy change is the sum of energy difference due to $N-1$ involved pairwise interactions between site $i$ and $j$, $\Delta E_{\text{tot}} \equiv \sum_j \Delta E_j$. The acceptance probability for the local update is,

$$P_{\text{Met}} = \exp\left(-\beta \left[\sum_j \Delta E_j\right]^+\right). \tag{5}$$

Despite the simple form of Eq. (5), implementing the Metropolis filter requires calculating the total energy change for $N-1$ interaction pairs, resulting in an expensive $\mathcal{O}(N)$ computational overhead. Consequently, long-range interactions can lead to significant performance issues, rendering the algorithm impractical for large-scale simulations.

This issue is even worse in the path-integral Monte Carlo (PIMC) methods when simulating long-range interacting quantum systems. PIMC methods involve mapping a $d$-dimensional quantum model onto a $(d + 1)$-dimensional classical system upon a specific expansion basis. The additional dimension is the imaginary-time $(\tau)$ direction, where continuous worldlines represent the state of each lattice site. In the path-integral formulation, the partition function of the quantum model can be seen as the weighted sum over all possible configurations in $(d + 1)$-dimensional space-time. By sampling these configurations, the PIMC method can accurately determine the thermodynamic properties of the quantum model.

Given an expansion basis, the Hamiltonian of a quantum model can be divided into a diagonal term and a non-diagonal term, $\mathcal{H} = \hat{K} + \hat{U}$. Consider a long-range interacting quantum model with $N$ site and pairwise long-range interactions in the diagonal term, $\mathcal{H} = \hat{K} + \sum_{i,j} \hat{U}_{ij}$. The probability weight of a configuration $\mathcal{S}$ can be expressed as:

$$W(\mathcal{S}) = K(\mathcal{S}) \exp[-U(\mathcal{S})]. \tag{6}$$

Here, $K(\mathcal{S})$ is the weight factor due to off-diagonal terms, and $U(\mathcal{S})$ is the total potential energy of long-range diagonal interactions,

$$U(\mathcal{S}) = \sum_{i,j} \int_0^\beta U_{ij}(\tau) d\tau. \tag{7}$$

$U_{ij}(\tau)$ is interaction energy between site $i$ and $j$ at imaginary-time $\tau$.

The Metropolis algorithm can be used in PIMC. Consider a local update $\mathcal{S} \to \mathcal{S}'$ that only changes the potential energy of the configuration. The state on the $i$-th site within a certain imaginary-time interval $[\tau_1, \tau_2]$ is modified. The Metropolis filter of this update is,

$$P_{\text{Met}} = \exp\left(-\left[\sum_j \Delta U_j\right]^+\right), \tag{8}$$

where $\Delta U_j = \int_{\tau_1}^{\tau_2}\left[U_{ij}^{\text{new}}(\tau) - U_{ij}^{\text{old}}(\tau)\right] d\tau$, is the energy change induced by the interaction between worldline $i$ and $j$ within the time interval $[\tau_1, \tau_2]$. As in the classical case, implementing Eq. (8) requires evaluating the total energy difference, which has a computational complexity of $\mathcal{O}(N)$. One must search for the states between $\tau_1$ and $\tau_2$ on worldlines that interact with the $i$-th site and perform $N-1$ integrations. However, the need for state searches and integrations makes this process more computationally demanding than the classical case. This computational complexity underscores the need for more efficient approaches to handling long-range systems in PIMC simulations to advance further our understanding of the behavior of many-body quantum systems.

## 2.2 Factorized Metropolis filter

Although using the Metropolis filter in various MCMC simulations has long been a conventional practice, physicists developed acceptance probability of other forms, such as the heat-bath algorithm [50]. A recent work by M. Manon et al. [51] introduces a new type of acceptance probability, named the *factorized Metropolis filter*, by factoring the Metropolis filter. It is the foundation of the event-chain Monte Carlo (ECMC) method [51–53], an irreversible and rejection-free MCMC algorithm. Instead of the detailed balance, the maximal global balance is fulfilled in this algorithm, where the probability flow between two configurations is unidirectional, and the flow back to the same configuration is forbidden. The factorized Metropolis filter offers a more flexible interpretation of the sampling process and opens up new possibilities for designing efficient MCMC algorithms.

In a long-range interacting classical system with $N$ sites, a local update on the $i$-th site is subject to the Metropolis filter described in Eq. (5). By factoring out the summation of pairwise energy changes, one obtains the factorized Metropolis filter for this update,

$$P_{\text{fac}} = \prod_j \exp\left(-\beta\left[\Delta E_j\right]^+\right) \equiv \prod_j P_j. \tag{9}$$

This acceptance probability, which is the product of independent factors $P_j \equiv \exp\left(-\beta\left[\Delta E_j\right]^+\right)$, also fulfills the detailed balance condition.

To determine the fate of a proposed update using the factorized Metropolis filter, one can straightforwardly compute the value of $P_{\text{fac}}$ and decide whether to accept the update based on it; however, this method requires exactly $N-1$ energy evaluations, which offers no advantages over the original Metropolis filter. Furthermore, it might result in a lower overall acceptance rate due to the lack of compensation between different $\Delta E_j$ terms.

Instead of considering Eq. (9) as a single trial with only acceptance or rejection, one can view the factorized filter as a series of $N-1$ independent trials with probability $P_j$. Factor $P_j$ is the probability of accepting the update by the energy change $\Delta E_j$ resulting from the interaction between site $i$ and $j$. A slightly cleverer method, as shown in algorithm 1, takes advantage of the independence of factors: for a proposed update, one performs sequential tests on all $P_j$ and rejects the update if any of the tests fail. The proposed update is accepted if and only if all the factors give permission, known as the consensus rule. This method requires more random numbers but allows for on-the-fly energy calculation of $\Delta E_j$. Since the first rejected factor will reject the entire update, the number of $\Delta E_j$ evaluated for rejection is less than or equal to $N-1$. Nevertheless, one must still compute all $\Delta E_j$ to accept an update, and the average complexity of this implementation remains $\mathcal{O}(N)$.

Although the factorized Metropolis filter does not immediately solve the computational complexity overhead, it provides a more flexible interpretation of the sampling process of an update's fate, which enables us to develop an efficient sampling scheme for long-range interacting systems.

## 2.3 Recursive clock sampling

In this subsection, we extend the clock sampling [34] to long-range interacting quantum models. The term *recursive clock sampling* is adopted to better elaborate the sampling process. This process is used to determine the fate of the attempted update, which substantially reduces the computational overhead arising from long-range interactions. Rather than employing the Metropolis filter with only binary outcomes (acceptance or rejection), the clock sampling scheme determines an update's fate using the factorized Metropolis filter by sampling from a probability distribution of clocks. These clocks describe the possible outcomes of the factorized Metropolis filter. They are efficiently sampled by formulating them into a tree-like structure, enabling the sampling process to be largely configuration-independent and circumventing costly energy evaluations.

In the remainder of this section, we elucidate the recursive clock sampling scheme for proposed updates within the PIMC framework. To simplify the explanation, let us consider a local update on the $i$-th worldline in a long-range interacting quantum system that only changes the configuration's diagonal potential energy. The acceptance probability of the update is governed by the factorized Metropolis filter,

$$P_{\text{fac}} = \prod_j \exp\left(-\left[\Delta U_j\right]^+\right) \equiv \prod_j P_j, \tag{10}$$

---

**Algorithm 1:** Factorized Metropolis filter

**for** $j = 1$ **to** $N-1$ **do**
    Evaluate $P_j$;
    **if** `ran` $>$ `P`$_j$ **then**
        **return** False;                     // Rejection
    **end**
**end**
**return** True;                         // Acceptance

---

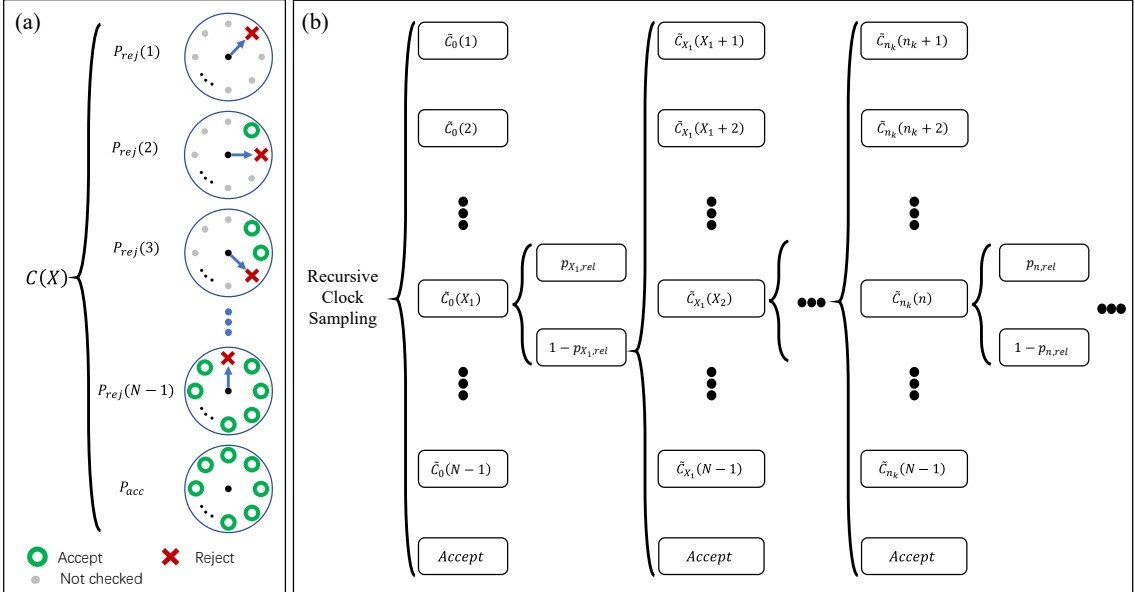

Figure 1: (a) In the clock sampling process, one determines the fate of a proposed update by sampling clocks from the probability distribution $C(X), X \in [0,N]$. Each clock represents a possible outcome of the factorized Metropolis filter. The first $N-1$ clocks are the first rejection events, and the clock hand points to the first rejecting factor. The last clock is the acceptance events, where all factors permit the update. (b) Schematic illustration of the recursive sampling process of the first-bound-rejection events on a tree structure.

where $P_j \equiv \exp\left(-\left[\Delta U_j\right]^+\right)$ is the $j$-th factor defined as the probability of the update being accepted by the $j$-th energy difference $\Delta U_j$. Here, $j = 1, 2, \ldots, N-1$ represents the indices of the neighboring worldlines that interact with the $i$-th worldline, and $\Delta U_j$ denotes the corresponding energy changes induced by the update.

The clock sampling scheme comprises two major components: firstly, the acceptance-rejection of an update is identified as a set of first-rejection events, and then a recursive sampling scheme is formulated to sample the probability distribution formed by these events efficiently.

*First-rejection events.* In order to map the acceptance-rejection of a proposed update to a set of events, we observe that Eq. (10) has a product form. Thus, $P_j$ can be seen as the probability of the successful outcome of an independent Bernoulli trial associated with the interaction between $i$ and $j$. In this context, a Bernoulli trial refers to a random experiment with two possible results: "acceptance" and "rejection." In other words, in the factorized Metropolis filter, each interaction can independently determine whether to accept or reject the update according to the corresponding $P_j$. Hence, instead of a single trial with probability $P_{\text{fac}}$, we can perform a sequence of $N-1$ independent trials, each with acceptance probability $P_j$, with in total $2^{N-1}$ possible outcomes. $P_{\text{fac}}$ can be defined as the probability of the *acceptance event* where all $N-1$ experiments give "acceptance". Meanwhile, the update is rejected if any of the experiments fail. Since the trials are performed sequentially, we can then define the *first-rejection event*, where the $X$-th factor in the factorized Metropolis filter is the first to reject the update. Once a first-reject event is identified, the update is rejected, regardless of the remaining trials. The probability of the first rejection event at the $j$-th factor is given by,

$$P_{\text{rej}}(j) = h_j \prod_{k=1}^{j-1}(1 - h_k). \tag{11}$$

Here, $h_j$ is the hazard rate of $P_{\text{rej}}(j)$ [54], and we identify the hazard rate $h_j \equiv 1 - P_j$ as the probability of the update being rejected by the $j$-th factor. Within this formulation, the probability of the acceptance event is,

$$P_{\text{acc}} = \prod_{k=1}^{N-1}(1-h_k).$$ (12)

The rejection and acceptance events can be clearly illustrated using the clocks in Fig. 1(a). The $j$-th index on the clock dial symbolizes the $j$-th factor $P_j$. The hand of a clock points to the first-rejecting factor, where all preceding factors permit the updates, and those following it are not checked. When there is no clock hand, all factors accept the update, and the clock represents the acceptance events. In this context, the term *clock* alludes to the potential outcomes of the factorized Metropolis filter. Instead of sequentially checking each factor, the clock sampling process aims to sample the probability distribution formed by these clocks directly:

$$\mathcal{C}(X) = \begin{cases} P_{\text{rej}}(X), & \text{if } 1 \le X \le N-1, \\ P_{\text{acc}}, & \text{if } X = N. \end{cases}$$ (13)

If the sampled clock alarms a first-rejecting event, then the update is rejected immediately, while if the acceptance clock is generated, the update will be directly accepted.

In conclusion, through the above mapping, we convert the sampling of factorized Metropolis filter in Eq. (10) into the task of sampling the discrete probability distribution $\mathcal{C}(X)$ of size $N$ with hazard rate $h_j$.

*The recursive clock sampling scheme.* The straightforward sampling scheme of distribution $\mathcal{C}(X)$ involves sequential tests of each hazard rate $h_j$. However, it is worth noting that the rejection probabilities $h_j$ for long-distance interactions decay algebraically with the system size, making rejections for long-range interactions very unlikely to occur. Additionally, as the system size increases, the leading term of $\mathcal{C}(X)$ also exhibits a power-law decay. This implies that first-rejection events are most likely to occur for interactions in the proximity of the updated worldline and there is no need to test for all factors in the tail. Instead, we can sample the distribution of $\mathcal{C}(X)$ directly.

Various methods exist for sampling a discrete probability distribution, such as the inversion method and Walker's alias method [55, 56]. However, these methods cannot be directly applied because $\mathcal{C}(X)$ is *configuration-dependent*, as the hazard rates $h_j$ are calculated from the configurations $S$ and $S'$, which vary during the MC simulation. Consequently, any method that requires the knowledge of all $N-1$ hazard rates will have at least $\mathcal{O}(N)$ complexity and will not be more efficient than the original Metropolis method.

To address this limitation and circumvent expensive energy evaluations, we demonstrate the recursive clock sampling process where configuration-independent distributions are sampled recursively to sample the target distribution of the clock. First, let us introduce a *configuration-independent* probability $\hat{h}_j \ge h_j$ for each factor, named *bound hazard rate*. This probability is determined by considering the "worst possible" local configuration that can lead to the largest energy change $\Delta \hat{U}_j$ after applying the update. A two-step process is used to determine whether a factor $j$ accepts the update. The first step is a *bound trial* with a rejection probability of $\hat{h}_j$. The outcome can be either bound acceptance or bound rejection. A bound acceptance means that the update is accepted in this trial for the worst case, and thus, it implies a true trial acceptance, with no need to examine the associated local configuration. In contrast, when a bound trial rejection occurs, one has to compute the actual and configuration-dependent rejection probability $h_j$ and sample the true rejection with relative probability,

$$p_{j,\text{rel}} = h_j/\hat{h}_j.$$ (14)

There are three potential outcomes at each factor $j$:

1. *Bound acceptance*: the update is accepted with $1-\hat{h}_j$.

2. *Relative acceptance*: the update is first bound rejected with $\hat{h}_j$ and then accepted with relative probability $1-p_{j,\text{rel}}$.

3. *True rejection*: the update is rejected with both $\hat{h}_j$ and $p_{j,\text{rel}}$.

Both bound acceptance and relative acceptance contribute to the overall acceptance of factor $j$, so the acceptance probability of factor $j$ is still $1-\hat{h}_j+\hat{h}_j(1-p_{j,\text{rel}})=1-h_j$. Meanwhile, the true-rejection event is equivalent to the original rejection event with probability, $\hat{h}_j\times p_{j,\text{rel}}=h_j$. Since the individual acceptance-rejection probability of each factor remains unchanged, one can conclude that introducing the bound hazard rate does not change the final fate of the update.

A vital characteristic of this two-step sampling scheme is that the hazard rate $h_j$ is evaluated when the update is bound rejected at factor $j$. Therefore, we can define a non-homogeneous Bernoulli process with hazard rate $\hat{h}_j$ to generate bound-rejection events and determine whether these factors truly reject the update. For a bound-rejection event at factor $j$, the corresponding relative probability is computed to test if this factor genuinely rejects the update. If it is not a true rejection event, i.e., the update is accepted with relative probability $1-p_{j,\text{rel}}$, the process has to continue to sample the next bound-rejection events.

Let us define $\tilde{\mathcal{C}}_{X'}(X)$ as the probability of the next bound-rejection event occurring at factor $X$ provided that the current bound-rejection event occurs at factor $X'$:

$$\tilde{\mathcal{C}}_{X'}(X) = \hat{h}_X \prod_{j=X'+1}^{X-1}(1-\hat{h}_j). \tag{15}$$

The corresponding bound-acceptance event is then,

$$\tilde{\mathcal{C}}_{X',\text{acc}} = \prod_{j=X'+1}^{N-1}(1-\hat{h}_j). \tag{16}$$

Similar to the first-rejection event case, these events form a probability distribution of size $N-X'$. By recursively sampling these distributions and the corresponding relative probability, one can efficiently sample the target distribution $\mathcal{C}(X)$.

As demonstrated in Fig. 1 (b), the recursive clock sampling scheme can be viewed as a sampling process on a tree structure. Starting at the first level, one generates a bound-rejection event at factor $X_1$ according to the *configuration-independent* distribution $\tilde{\mathcal{C}}_0(X_1)$ and performs the rejection test with probability $p_{X_1,\text{rel}}$. If factor $X_1$ does not truly reject the update, one goes to the next level and generates the next bound-rejection event relative to $X_1$. This process is recursively performed, generating a series of bound-rejection events at factor $\{X_1,X_2,X_3,\dots\}$, and until the first actual rejection occurs at specific $X_{\text{rej}}$ or the update is accepted by all $P_j$. The bound rejection does not change the actual rejection probability at each factor; therefore, this sampling scheme yields the same probability distribution for the first-rejection event $\mathcal{C}(X)$. At each level, the energy evaluation is performed only once, making the computation complexity $C$ the average number of levels during the sampling process. We define the bound consensus probability $P_B = \prod(1-\hat{h}_j)$ as in Ref. [34], and the complexity scales as $C \sim \mathcal{O}(\ln P_B/\ln P_{\text{fac}})$. If the bound consensus probability $P_B$ scales with $N$ as $P_{\text{fac}}$, the clock sampling scheme has a computational complexity of $\mathcal{O}(1)$. This scaling depends on the energy profile of the updates and, for a local update between $\tau_1$ and $\tau_2$ can be expressed as: $C \sim (\sum_i \max|\Delta E_i(\tau_1-\tau_2)|)/(\sum_i|\Delta E_i(\tau_1-\tau_2)|)$. Thus, the average complexity scales as

---

**Algorithm 2:** Recursive clock sampling scheme

$j \leftarrow 1$;
**while** $j \leq N$ **do**
    Generate the next bound-rejection event at $j'$ according to Eq. (15);
    $j \leftarrow j'$;
    **if** `ran` $< p_{j,rel}$ **then**
        **return** *Reject*;                                `// Rejection`
    **end**
**end**
**return** *Accept*;                                        `// Acceptance`

---

$\mathcal{O}(1)$ if both the numerator and the denominator scale similarly with $N$. For extensive system, both $\sum_i \max|\Delta E_i|$ and $\sum_i(|\Delta E_i|)$ converges to finite values, allowing for $\mathcal{O}(1)$ complexity per update. For sub-extensive, we expect the complexity to scale as $\mathcal{O}(N^\alpha)$ with $0 < \alpha < 1$, similar to classical cases [34]. Moreover, $\tilde{\mathcal{C}}_{X'}(X)$ is a configuration-independent distribution at each level, and several techniques exist to sample it efficiently. Consequently, the clock sampling scheme substantially reduces the computational complexity of long-range interactions.

*Off-diagonal weights and general proposal probabilities.* In the preceding discussion, we focus on a simple scenario where the proposed update only changes the diagonal long-range interaction term of the configuration weight, assuming a symmetrical proposal distribution. However, in the path-integral representation, it is essential for an ergodic update scheme to modify off-diagonal terms of the configuration as well. Furthermore, the proposal probabilities of updates are typically asymmetrical and non-trivial. Therefore, it is crucial to generalize the clock sampling to accommodate such cases.

Without loss of generality, let's consider an update that changes the off-diagonal terms of the configuration weight, $K(\mathcal{S}) \to K(\mathcal{S}')$ and has a proposal distribution $\mathcal{A}(\mathcal{S} \to \mathcal{S}')$. The acceptance probability of such an update is given by,

$$P_{\text{M-H}} = \min\left(1, \frac{\mathcal{A}(\mathcal{S}' \to \mathcal{S})K(\mathcal{S}')}{\mathcal{A}(\mathcal{S} \to \mathcal{S}')K(\mathcal{S})} \exp(-\Delta U)\right), \tag{17}$$

with $\Delta U = \sum_j U_j$. Therefore, by further factoring out the proposal probabilities and the off-diagonal weights, we obtain the factorized filter:

$$P_{\text{fac}} = P_{\mathcal{A}} \prod_{j=1}^{N-1} P_j. \tag{18}$$

In this factorization, an additional factor $P_{\mathcal{A}}$ is introduced to account for the off-diagonal weights and the proposal distribution of the update, which is given by,

$$P_{\mathcal{A}} = \min\left(1, \frac{\mathcal{A}(\mathcal{S}' \to \mathcal{S})K(\mathcal{S}')}{\mathcal{A}(\mathcal{S} \to \mathcal{S}')K(\mathcal{S})}\right). \tag{19}$$

Furthermore, the factor $P_{\mathcal{A}}$ can be formulated with great flexibility. One can incorporate the local diagonal terms of the Hamiltonian into $P_{\mathcal{A}}$, such as on-site potentials, so that $P_{\mathcal{A}}$ resembles the original acceptance probability excluding the energy changes due to long-range interactions.

It can be challenging to determine a configuration-independent bound hazard rate $\hat{h}_{\mathcal{A}}$ for $P_{\mathcal{A}}$ since it relies on the specific details of the update scheme. One possible approach to address this issue is to conduct an initial trial with acceptance probability $P_{\mathcal{A}}$ at the beginning of the

clock sampling. If this preliminary trial fails, the update is rejected immediately. Otherwise, one proceeds to generate bound rejection events for $P_j$ factors. This strategy effectively treats $P_{\mathcal{A}}$ as the first factor in the sampling process and sets $\hat{h}_{\mathcal{A}} = 1$. By employing this strategy, the clock sampling can be seamlessly integrated with different update schemes, thereby enhancing the overall efficiency of the algorithm.

*Box Technique.* A side effect of using a factorized Metropolis filter is that the overall acceptance probability may decrease due to factorization. This can be observed from the following inequality:

$$\left[\sum_j \Delta U_j\right]^+ \le \sum_j \left[\Delta U_j\right]^+ . \tag{20}$$

As a result, the overall acceptance probability of the factorized Metropolis filter is always less than that of the Metropolis filter. However, this is not a problem in most cases, except in glassy systems where $\Delta U_j$ can cancel each other dramatically. In such situations, the *box technique* can help alleviate the problem. The boxing technique takes advantage of the fact that the factorized Metropolis filter can be constructed with considerable flexibility: each factor $P_j$ may contain an arbitrary number of interactions. For instance, interactions can be grouped into $N_b$ boxes with tunable sizes $B_b$, and the filter becomes:

$$P_{\text{fac}}^{\text{Box}} = \prod_{b=1}^{N_b} \exp\left(-\left[\sum_{j=1}^{B_b} \Delta U_j\right]^+\right) . \tag{21}$$

When $N_b = 1$, the factorized Metropolis filter reduces to the original Metropolis filter since all interactions are in a single factor. The detailed balance condition will always be satisfied regardless. This leads to new optimization possibilities, which can be particularly useful in the case of glassy systems.

In summary, the recursive clock sampling process is an efficient sampling scheme to determine the fate of an attempted update in a long-range quantum system. It offers three major benefits: (i) *Reduced computational complexity*: The clock sampling process dramatically reduces the computational complexity per update from $\mathcal{O}(N)$ to $\mathcal{O}(N^\kappa)$ ($0 \le \kappa \le 1$). In most cases, $\mathcal{O}(1)$ update complexity can be achieved. (ii) *Flexible update scheme*: the clock sampling process is not limited to any specific update scheme. It can be integrated with various update strategies to enhance algorithm performance. (iii) *Box technique*: the clock sampling process can be constructed in various ways, enabling further optimization for specific models. The interactions in the Hamiltonian can be grouped into boxes of tunable sizes to increase the overall acceptance rate. By reducing the computational complexity of the Metropolis filter's long-range interaction terms, the proposed clock sampling scheme allows for the efficient exploration of a diverse array of fascinating physical phenomena in long-range interacting systems.

## 3 Efficient implementation of recursive clock sampling

This section delves into the implementation of the recursive clock sampling scheme. Specifically, we focus on efficiently generating the bound-rejection events from a probability theory perspective. As discussed in the previous section, the recursive clock sampling process relies on recursively sampling a tree structure of bound-rejection events, significantly reducing computational complexity. At each iteration, one generates the next bound-rejection event at factor $X$ according to the configuration-independent distribution given by Eq. (15). Hence, to obtain

an optimized implementation of the clock sampling scheme, we seek an efficient and robust method capable of generating these events.

In the context of probability theory, this is the famous problem of discrete random variate generation, which has been studied for many years [54,57]. A discrete random variate $X$ takes only integer values in a finite set, such as $k \in 1, 2, \ldots, n$. Its distribution follows the probability mass function (PMF) denoted as $p(k) = P(X = k)$, where $P(X = k)$ is the probability of $X$ taking the value $k$. In the subsequent discussion of this section, we define $X$ as a discrete random variable that describes the next bound-rejection events, with its value being indices of the factor where the next bound-rejection occurs, and its corresponding PMF $p(k)$ satisfies Eq. (15).

Various algorithms exist to sample discrete random variates. However, $p(k)$ exhibits two special intrinsic features. First, $p(k)$ changes during the simulation to ensure optimal performance. Although $p(k)$ is configuration-independent, the bound hazard rate should be chosen based on the detail of the update, such as the update's range in the $\tau$-direction. In addition, the distribution of bound rejection events is also different at each level of a clock sampling process. Secondly, $p(k)$ is a distribution whose probability is not known explicitly. For a given update, $\hat{h}_j$ can be directly computed for any index $j$, while the probability of a particular bound-rejection event is difficult to calculate. We identify $\hat{h}_j$ as the hazard rate function of distribution $p(k)$ from the definition. Thus, $p(k)$ is a distribution with known hazard rates. When sampling $p(k)$, these two properties must be considered.

This section briefly introduces a class of algorithms suitable for sampling $p(k)$, named the thinning methods. Lastly, we thoroughly explain our implementation of the clock sampling scheme using the thinning method and provide pseudocode for added clarity.

*Thinning method.* The bound rejection event is described by a distribution $p(k)$ with known hazard rate $\hat{h}_k$,

$$p(k) = \begin{cases} \hat{h}_k \prod_{j}^{k-1} \left(1 - \hat{h}_j\right), & \text{if } k \in [1, N-1], \\ \prod_{j=1}^{N-1} \left(1 - \hat{h}_j\right), & \text{if } k = N. \end{cases} \tag{22}$$

A straightforward algorithm to sample the above distribution is the sequential test method [36, 54]. One starts from $k = 0$ and sequentially tests if the random variable can take the values $0, 1, 2, \ldots, N$. It is equivalent to a series of non-homogeneous Bernoulli trials with failure probability $\hat{h}_k$. Similar to the inversion method by sequential search, this method has a time complexity of $\mathcal{O}(N)$. However, the sequential test method requires one uniform random variable per iteration.

In 1985, Shanthikumar observed that for discrete hazard rates $\hat{h}_k$ with supremum $\rho < 1$, the sequential test method can be accelerated by jumping ahead more than 1 in each iteration. Based on this observation, the *discrete thinning method* is proposed [36]. The method's basic idea is to generate a sample from a distribution with a dominating rate $g_k \geq \hat{h}_k$ and then thin it down to the desired distribution by rejecting some of the events.

Consider a constant dominating rate $g_k = \rho$, for all $\hat{h}_k \leq \rho$. Such a dominating distribution is simply a geometric distribution with parameter $p = \rho$, which can be easily generated using the inversion method, described in A.1. The discrete thinning method works as follows: one starts with $X \leftarrow 0$. At every iteration, one generates a geometric distributed random number $k$, updating the value $X \leftarrow X + k$, and then rejects the event with probability $\hat{h}_X/\rho$. This process repeats until a sample $X$ is accepted. The resulting random number $X$ follows the target distribution. The expected number of iterations for the discrete thinning method is $\rho E(X)$ since the average jump size is $1/\rho$. The method reduces to the sequential test method in the $\rho = 1$ limit. Consequently, when sampling a given distribution, the smaller $\rho$, the more dramatic the improvement. Therefore, the discrete thinning method can be advantageous in clock sampling where only the hazard rate of the bound rejection events $\hat{h}_j$ is known.

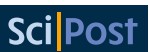

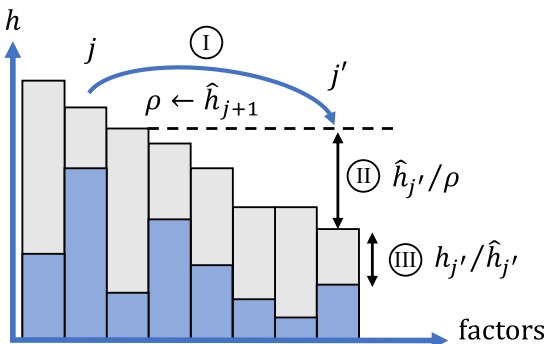

Figure 2: A schematic diagram of one level of the clock sampling. The blue box represents the hazard rate $h_j$ of factors, and the gray box represents the corresponding bound hazard rate $\hat{h}_j$. Starting from the $j$-th factor, the first step (I) generates a jump to the $j'$-th factor using a geometric random number with parameter $\rho$. The second step (II) is to accept $j'$ as a bound-rejection event with relative probability $\hat{h}_{j'}/\rho$. If $j'$ is rejected, then one goes back to (I). Otherwise, if $j'$ is indeed a bound-rejection event, then one goes to the third step (III) to check if factor $j'$ truly rejects the update with $h_{j'}/\hat{h}_{j'}$.

In the clock sampling, we are interested in whether a given update is eventually accepted. Thus, the order of factors in Eq. 9 is irrelevant. One can sort the factors by their bound hazard rate, such that $\hat{h}_j$ is decreasing. Then, the new distribution has a decreasing hazard rate, referred to as a DHR distribution, which can be initialized before the actual simulation. The performance of the thinning method for a DHR distribution can be further improved by dynamically lowering the constant dominating rate $\rho$. This method is formally named the *dynamic thinning method* [36]. For the bound rejection events that follow a discrete distribution $p(k)$ with decreasing hazard rate, $\hat{h}_0 > \hat{h}_1 > \cdots \hat{h}_{N-1}$, one starts with $X \leftarrow 0$. At every iteration, one generates a geometrically distributed random number $k$ and updates the value $X \leftarrow X + k$. Then, one attempts to accept this value with probability $\hat{h}_X/\rho$. If so, a sample is successfully generated. Otherwise, the upper bound $\rho$ is lowered to equal the hazard rate value of the subsequent factor $\hat{h}_{X+1}$. The process repeats until a sample $X$ is accepted. Therefore, the dynamical thinning method allows for larger jump sizes in the tail of the DHR distribution, thereby improving the sampling process's performance.

The bound hazard rates $\hat{h}_j$ are generally very small except for those corresponding to short-range interactions because the value of $\hat{h}_j$ depends on the strength of the corresponding long-range interaction, which decays algebraically with the distance. This property makes the bound rejection event hardly occur for interactions in the tail of the distribution. More importantly, it implies that the distribution has a long but small tail, where the dominating rate $\rho$ of the dynamic thinning method can also be very small, ensuring the high efficiency of the algorithm.

Furthermore, the dynamic thinning method can compute $\hat{h}_j$ on the fly, provided that the order of $\hat{h}_j$ is known in advance. Therefore, if one can select a sequence of $\hat{h}_j$ whose order remains constant throughout the simulation, it is necessary to sort the $\hat{h}_j$ only once before the actual simulation. This order can then be stored and used in the dynamic thinning method, thereby eliminating the need for additional initialization procedures for different values of $\hat{h}_j$.

In conclusion, given its high efficiency and streamlined operations, the dynamic thinning method is an optimal choice for generating bound rejection events within the clock sampling scheme.

---

**Algorithm 3:** Clock sampling with dynamic thinning

**Input:** A proposed update $\mathcal{S} \to \mathcal{S}'$
**Output:** The update is accepted or rejected
**Initialization**: Identify and reorder the bound hazard rates of all factors
$\hat{h}_1 > \hat{h}_2 > \cdots > \hat{h}_{N-1}$.

$j \leftarrow 0$;
**while** $j < N$ **do**
    $\rho \leftarrow \hat{h}_{j+1}$;
    Generate random variate $u \in [0, 1)$;
    $j \leftarrow j + \lceil \frac{\log(u)}{\log(1-\rho)} \rceil$;
    **if** $j \geq N$ **then**
        **break**
    **end**
    Generate random variate $v \in [0, 1)$;
    **if** $v < \frac{\hat{h}_j}{\rho}$ **then**
        Evaluate $h_j = 1 - P_j$;
        Generate random variate $w \in [0, 1)$;
        **if** $w < p_{j,\mathrm{rel}}$ **then**
            **return** False;
        **end**
    **end**
**end**
**return** True;

---

*Implementation of recursive clock sampling.* We demonstrate one possible implementation of recursive clock sampling using the dynamic thinning technique to generate the bound-rejection events. The pseudocode is given in Alg. 3, and the schematic diagram is shown in Fig. 2. For a long-range interacting system of size $N$, one first identifies and reorders the bound hazard rates $\hat{h}_j$ of all factors, denoted as $\hat{h}_1 > \hat{h}_2 > \cdots > \hat{h}_{N-1}$. The bound hazard rates are selected based on the properties of the model to be studied. To determine the fate of a proposed update $\mathcal{S} \to \mathcal{S}'$, one starts with $j \leftarrow 0$. One increments $j$ via a geometric random number with parameter $\rho = \hat{h}_{j+1}$

$$j \leftarrow j + \left\lceil \frac{\log(\texttt{ran})}{\log(1 - \hat{h}_{j+1})} \right\rceil, \tag{23}$$

where $u \equiv \texttt{ran}$ is a uniform random variable, and $\lceil x \rceil$ is the ceiling function that returns the smallest integer larger than or equal to $x$. One then tests if this new $j$ is truly a bound rejection event with probability $\hat{h}_j/\rho$. One repeats this process until a bound rejection event is successfully generated at $j$-th factor. The next step is to check whether the bound rejection is an actual rejection with probability $p_{j,\mathrm{rel}} = h_j/\hat{h}_j$. In this step, the energy difference is evaluated to obtain $h_j$. The sampling terminates when a true rejection is found; otherwise, one goes to the next level and generates new bound rejection events. The process continues until the update is accepted, which occurs when $j \geq N$.

The algorithm integrates the dynamic thinning method and the clock sampling scheme for a proposed update. To initialize the algorithm, one needs to store the order of $\hat{h}_j$, which can be determined before the simulation begins. This approach is both straightforward and efficient, making it ideal for large-scale simulations of long-range interacting systems.

# 4 Clock factorized quantum Monte Carlo algorithms

In this section, we introduce a class of Monte Carlo algorithms that utilize clock sampling to determine the fate of an attempted update, which we call the clock factorized quantum Monte Carlo (clock factorized QMC) method. Specifically, we demonstrate three different clock factorized QMC algorithms in the path-integral formulation to simulate typical quantum systems with long-range interaction in condensed matter physics. Firstly, we designed a clock factorized Metropolis algorithm that employs a local Metropolis-type update scheme to simulate the long-range transverse field Ising model (LRTFIM). Secondly, integrating the clock sampling with the worm update, we develop a clock factorized worm algorithm to simulate the extended Bose-Hubbard model (EBHM). Finally, we enhanced the clock factorized worm algorithm using additional efficient long-range hopping updates. We utilized this improved algorithm to simulate the long-range XXZ Heisenberg model (LRXXZ) by first mapping the model to a hardcore Bose-Hubbard model with both long-range density-density interaction and long-range hopping.

When constructing a clock factorized QMC algorithm, careful consideration must be given to two crucial elements. The first element is the box technique introduced in the previous section, where long-range interaction terms are grouped into boxes to increase the overall acceptance rate. This study does not cover systems with glassy long-range interactions where the box technique can significantly affect the algorithm's performance, so we set the box size to 1 for simplicity, i.e., each factor contains only one pairwise interaction. The second element is the proper choice of the bound hazard rate, denoted as $\hat{h}_j$. As previously discussed, the value of $\hat{h}_j$ governs the average step size of the clock sampling, thus significantly affecting the algorithm's performance. However, once these steps have been completed, the design and implementation of the clock factorized QMC algorithm for a given model is typically straightforward. The approach involves selecting a state-of-the-art update scheme for the model and integrating the clock sampling process with the updates. This implementation process requires only minimal modifications of an existing code by replacing the Metropolis filter of the original algorithm with a clock sampling step, while the proposal of updates and the actual update operations remain unchanged. Therefore, in the following description of clock factorized QMC algorithms, we shall focus on the vital ingredients of a clock factorized QMC algorithm, such as deriving an expression for bound hazard rate $\hat{h}_j$, while we only briefly describe the update schemes without diving into the details.

To assess the efficiency of the clock factorized QMC algorithm in reducing computational effort, we define the average computational complexity of the local update, $C$, as the number of energy evaluations required for each local update:

$$C = \#\text{pairwise energy evaluations per local update.} \tag{24}$$

The expectation value of $C$ provides a direct and quantitative measure of the computational effort demanded by the algorithm, independent of the hardware configuration. Because pairwise energy evaluations typically represent the most computationally intensive step in simulating long-range interacting systems, this definition offers a meaningful proxy for the runtime per Monte Carlo step. This is verified in the performance benchmark presented later. For the conventional Metropolis filter, $C = N - 1$, while the clock factorized QMC algorithms have substantially lower $C$. In the following sections, we refer to the average computational complexity of the local update as *Complexity* for brevity. Since the primary goal of this work is to introduce and validate the clock factorized QMC methodology, and further optimizations of both update strategies and implementations for specific physical systems and parameters are available, we focus on the complexity of the algorithm in this study. A detailed benchmark of physical observables is left for future work, which involves studying specific physical systems.

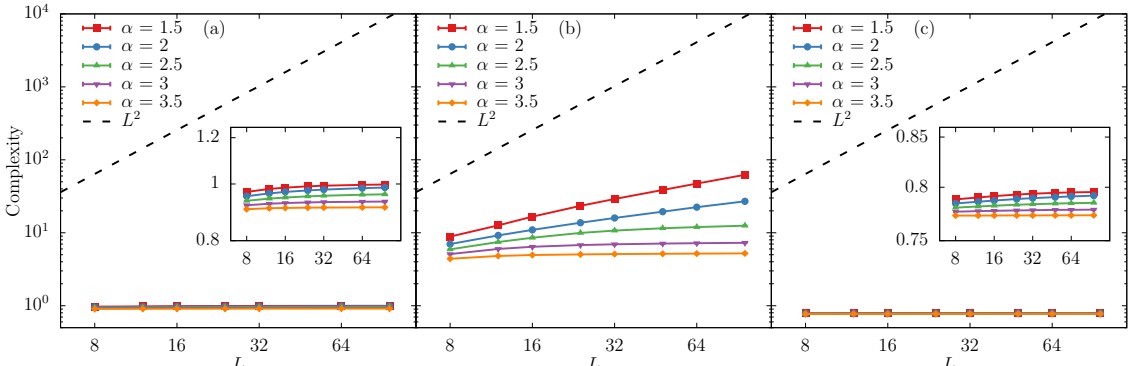

Figure 3: Average complexity of Monte Carlo update of 2D models. (a) long-range transverse field Ising model using Metropolis update, (b) extended Bose-Hubbard model using worm update, (c) long-range XXZ model using worm update with long-range hopping.

Simulations of these models are performed on both 2D square lattices and 3D cubic lattices of various sizes, represented as $L$. The complexities of the new algorithm for each model are shown in Fig. 3 and 4.

We also conduct several controlled performance benchmarks, to demonstrate the advantages of the new algorithm. To ensure consistency and minimize variations in performance measurements, these benchmarks are executed on a uniform hardware setup with an Intel Core i7-12700K CPU and 16 GB of DDR4-3200 dual-channel RAM. The evaluation focused not only on computational complexity but also entailed a direct comparison between the computation time per sweep, denoted as $\tau$, and the acceptance ratio. These quantities are compared for these models using the proposed clock factorized QMC algorithms and the algorithms with the conventional Metropolis filter. The benchmark results are presented in Fig. 5, 6 and 7. The results suggest that the new algorithms provide an efficient approach to large-scale simulation of long-range interacting systems. They allow accurate investigation of the physical properties of 3D long-range quantum models, which was previously hindered by substantial computational demands.

## 4.1 Clock factorized Metropolis algorithm

The transverse field Ising model (TFIM) is one of the most famous quantum spin models. The competition between ferromagnetic spin exchange interaction and transverse field can lead to rich physics [11,24]. It has been studied extensively using various numerical methods, such as quantum Monte Carlo and density matrix renormalization group [58]. It serves as a simplified model for many physical systems, including neutral atom array [59,60] and superconducting qubits [61].

In contrast to the conventional TFIM, in the long-range transverse field Ising model, the interactions between Ising spins are not restricted to nearest-neighbor pairs; instead, there is a power-law decay of the coupling strength with distance. The Hamiltonian of the long-range transverse field Ising model (LRTFIM) is given by,

$$\mathcal{H} = -\sum_{i,j} \frac{J}{r_{ij}^{\alpha}} \sigma_i^z \sigma_j^z - h \sum_{i=1}^{N} \sigma_i^x \,. \tag{25}$$

Here, $J > 0$ is the ferromagnetic coupling strength along the $z$-direction, and the power $\alpha$ determines the range of interactions between spins. The summation $\sum_{i,j}$ is over all pairs of

spins $i$ and $j$ on the lattice. The symbols $\sigma_i^z$ and $\sigma_i^x$ are Pauli matrices acting the $i$-th Ising spin, $h$ is the transverse magnetic field strength, and $N$ is the total number of spins in the system. The model reduces to the nearest-neighbor model in the limit $\alpha \to \infty$, while in the limit $\alpha \to 0$, all spins are coupled equally, and the model is a transverse field Ising model on a complete graph.

For the path-integral formulation of LRTFIM, we choose the spin state in $z$-direction as the basis, $|\sigma_1, \sigma_2, \ldots\rangle$, where $\sigma_i = \pm 1$ represents the up/down spin state on the $i$-th site. The configuration of the LRTFIM consists of $N$ worldlines made of segments. Each segment represents an imaginary time interval where the spin state remains unchanged, and the interface between two different segments is called a *cut*. When there is only one segment on a worldline, the segment can be considered as a ring without any cuts. In this expansion basis, the statistical weight of a configuration $\mathcal{S}$ is given by,

$$W_{\mathcal{S}} = \left( \prod_{k=1}^{\mathcal{N}} d\tau_k \right) h^{\mathcal{N}} \exp \left\{ \sum_{i,j} \int_0^\beta \frac{J}{r_{ij}^\alpha} \sigma_i(\tau) \sigma_j(\tau) d\tau \right\}, \tag{26}$$

where $\mathcal{N}$ is the number of cuts, and $\sigma_i(\tau)$ is the spin state at a space-time point $(i; \tau)$. The state of a worldline flips at imaginary time $\tau_k$ with $(k = 1, \ldots, \mathcal{N})$.

We employ a standard Metropolis-type update scheme for LRTFIM. The term "Metropolis-type" means that the update operations are local, i.e., modify only one segment at each MC step. This update scheme consists of two pairs of operations. (a) *Create/delete segment*. The first pair of operations manipulates the configuration by inserting a new segment or deleting an existing segment. To create a new segment, one randomly picks an existing segment from the configuration and then flips the spin state between the two uniformly chosen points in the segment. Conversely, the "delete segment" update is the reverse process of the "create segment" update. This procedure randomly chooses an existing segment and flips its spin state to remove it from the configuration. These operations change the number of segments in the configuration. (b) *Move cut*. The second operation moves the temporal location of an existing cut without altering the number of segments. To do this, one randomly chooses a cut and shifts it to a new position in the range bounded by its next and previous cuts. The move segment operation is its own reverse process. Using these local update operations, we can efficiently explore the configuration space of the long-range Ising model. These operations are then combined with the clock sampling process to obtain the clock factorized Metropolis algorithm. In this update scheme, both operations are local updates that modify the spin state within an imaginary time interval during which the spin state remains constant. Hence, it is possible to consider an update that flips a segment between $\tau_1$ and $\tau_2$ on the $i$-th site, and the initial spin state in this interval is represented by $\sigma_i$. The factorized Metropolis filter of this update is $P_{fac} = P_{\mathcal{A}} \prod_j P_j$. Here, $P_{\mathcal{A}}$ is a factor that depends on the detail of an update, as discussed in Section 2. Here, we take the create segment operation as an example:

$$P_{\mathcal{A}}^{\text{crea}} = \min \left\{ 1, \frac{N_{\text{seg}} h^2}{N'_{\text{seg}} u(\tau_1, \tau_2)} \right\}, \tag{27}$$

where $N_{\text{seg}}$ ($N'_{\text{seg}}$) is the number of segments before (after) the creation of a new segment. Imaginary time positions $\tau_1$, $\tau_2$ are chosen with the uniform probability density $u(\tau_1, \tau_2) = 2/(\tau_{\max} - \tau_{\min})^2$, where $\tau_{\max}$ ($\tau_{\min}$) is the starting (ending) time of the selected segment. On the other hand, the factors $P_j$, which are the key components of clock sampling, have a general form,

$$P_j = \exp \left\{ -\left[ 2J_{ij} \sigma_i \int_{\tau_1}^{\tau_2} \sigma_j(\tau) d\tau \right]^+ \right\}. \tag{28}$$

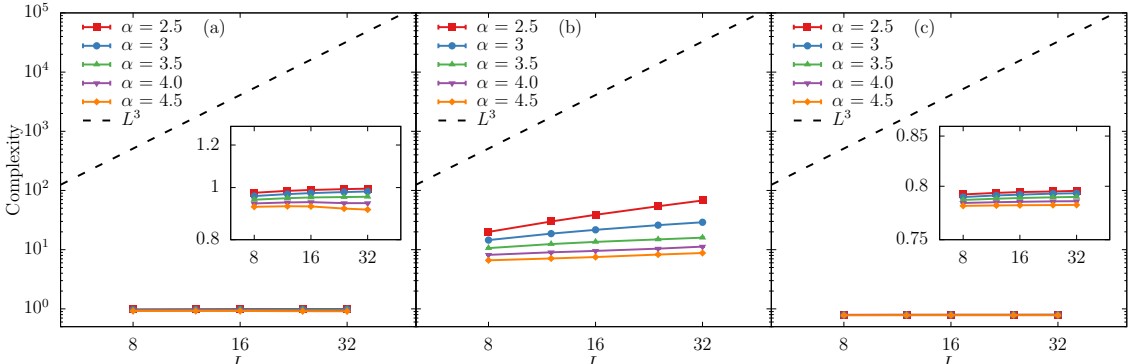

Figure 4: Average complexity of Monte Carlo update of 3D models. (a) long-range transverse field Ising model using Metropolis update, (b) extended Bose-Hubbard model using worm update, (c) long-range XXZ model using worm update with long-range hopping.

Here, $J_{ij}$ is the interaction strength between spins $i$ and $j$ given by $J_{ij} = J/r_{ij}^{\alpha}$.

To derive the bound hazard rate of $P_j$, one should first identify the factor's "worst background". In this context, the term "background" refers to the portion of unchanged configuration that interacts with the segment to be updated. In this example, the background is the spin state between $\tau_1$ and $\tau_2$ on the $j$-th worldline, represented by $\sigma_j(\tau)$ with $\tau \in [\tau_1, \tau_2]$. Hence, the "worst background" refers to a certain possible formation of background that can induce the most significant energy change after the update. This worst possible background depends solely on the characteristics of the model to be studied, thus making the bound hazard rate $\hat{h}_j$ independent of the actual configuration. In the LRTFIM, $\sigma$ take the value of $\pm 1$ and $J_{ij}$ is positive; thus, the worst background of $P_j$ is that case where the state between $\tau_1$ and $\tau_2$ on $j$ is same to that on the $i$-th worldline: $\sigma_j(\tau) = \sigma_i$ for $\tau \in [\tau_1, \tau_2]$. Consequently, the largest possible energy change is $2J_{ij}|\tau_2 - \tau_1|$, and the bound hazard rate is given by,

$$\hat{h}_j = 1 - \exp\left(-2J_{ij}|\tau_2 - \tau_1|\right). \tag{29}$$

It is evident that $\hat{h}_j$ has a configuration-independent expression and can be adopted in the clock sampling process.

The clock sampling method also requires that the bound hazard rate for an update must be arranged in decreasing order. This is achieved by computing all $N-1$ interaction strengths $J_{ij}$ for the $i$-th site at the beginning of the simulation, sorting them in decreasing order, and then using this sorted list for all updates. For a given local update, the value of $|\tau_2 - \tau_1|$ is constant, resulting in $\hat{h}_j$ being a function of the interaction strength $J_{ij}$. By using the sorted list of interaction strengths, the bound hazard rate is automatically ordered for any update, eliminating the need to explicitly sort $\hat{h}_j$ for each update. This approach ensures that the bound hazard rate is efficiently evaluated and arranged, meeting the requirement of clock sampling.

Simulations with various exponents of the long-range interaction and system sizes are conducted to comprehensively test the efficiency and robustness of the clock factorized Metropolis algorithm. The computational complexities of the long-range transverse field model for different exponents are compared, and the complexities of the clock factorized Metropolis algorithm of the LRTFIM on both 2D square and 3D cubic lattices are shown in Fig. 3(a) and Fig. 4(a), respectively. The simulations are conducted near the critical point of the corresponding short-range model, $h = 3.04433$ for the 2D square lattice [62, 63] and $h = 5.158129$ for the 3D cubic lattice [63]. The inverse temperature is fixed at $\beta = 10$. The almost constant computational complexity observed for different system sizes demonstrates a significant improvement

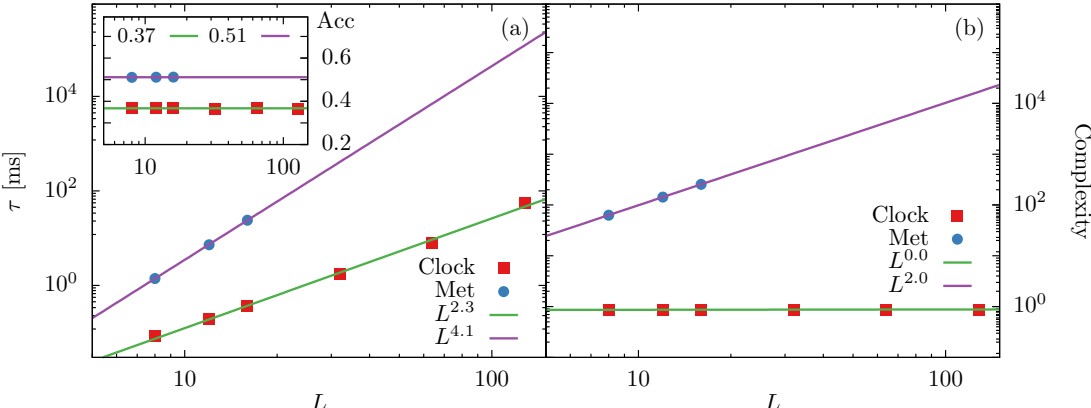

Figure 5: Performance benchmark of the clock factorized Metropolis algorithms compared with the conventional algorithm with Metropolis filter for 2D LRTFIM at $\alpha = 4.0$, $h = 5.2011$ and $\beta = 10$. Panel (a) shows the average time per sweep $\tau$ in milliseconds (ms) for both algorithms: $\tau$ of conventional algorithms scales approximately as $L^4$, while $\tau$ of clock factorized QMC algorithm scales as $L^2$. Notably, the CPU time per sweep for $L = 128$ by the clock factorized algorithm is comparable to that for $L = 16$ by the conventional Metropolis scheme. The inset presents the average acceptance ratio. Compared with the Metropolis scheme, the acceptance ratio of the new algorithm drops by a constant ratio $\gamma \approx 20\%$, from 0.51 to 0.37. Panel (b) displays the computational complexity per update for each case. The clock factorized Metropolis algorithm exhibits a $\mathcal{O}(1)$ computational complexity in contrast to the $\mathcal{O}(N)$ complexity of the conventional QMC algorithm.

in simulation efficiency achieved by the clock sampling algorithm. In Fig. 5, we present the result of performance benchmarks on the LRTFIM on a 2D square lattice. The model parameters are set at $\alpha = 4.0$ and $\beta = 10$, with $h = 5.2011$, which is near the critical point of the model [24]. The result demonstrates a near $\mathcal{O}(N)$ reduction of time per sweep for the clock factorized QMC algorithm compared with the conventional Metropolis scheme, approximately in the same order as the reduction of the computational complexity per update. In addition, the acceptance ratios of both algorithms do not show noticeable size dependence. Compared with the conventional scheme, the acceptance ratio of the new algorithm decreased by a constant ratio $\gamma \approx 27\%$, from $P_{\text{acc}} \approx 0.51$ to 0.37, thus the overall autocorrelation of the new algorithm increases by 27% because the two algorithms have identical physical dynamics [34]. Therefore, despite the slight increase of autocorrelation time for the clock factorized Metropolis algorithm, the overall improvement of update efficiency is $\mathcal{O}(N)$.

## 4.2 Clock factorized worm algorithm

The extended Bose-Hubbard model provides a fundamental framework for understanding the behavior of interacting bosonic particles in a lattice, making it highly relevant to AMO experimental setups. The model considers a system of bosonic particles that are confined to a lattice and interact with each other, where the interaction can be both short-range and long-range. The extended Bose-Hubbard model has been extensively studied in both theoretical [64–72] and experimental settings [73–79], with particular attention paid to the effects of long-range interactions due to their relevance in ultracold experiments.

The Hamiltonian of the EBHM is given by:

$$\mathcal{H} = -t \sum_{\langle i,j \rangle} \left( b_i^\dagger b_j + h.c. \right) + V \sum_{i<j} \frac{1}{r_{ij}^\alpha} n_i n_j + \frac{U}{2} \sum_i (n_i - 1) n_i + \sum_i \mu n_i . \tag{30}$$

Here, $b_i^\dagger$ ($b_i$) is the bosonic creation (annihilation) operator on $i$-th site, and $n_i \equiv b_i^\dagger b_i$ is the bosonic particle number operator. The Hamiltonian is a sum of several terms. The first term describes the nearest-neighbor hopping of bosons, where $t$ is the hopping strength. The second term sums over all pairwise long-range density-density interactions, controlled by the interaction strength $V$ and an exponent $\alpha$. $r_{ij}$ is the distance between $i$-th and $j$-th sites. The third term is the on-site repulsion with strength $U$, and the fourth term controls the filling fraction via the chemical potential $\mu$.

One of the state-of-the-art methods for simulating the extended Bose-Hubbard model is the worm algorithm, which is a highly successful PIMC algorithm for studying systems without the sign problem [21,29,80]. It is based on the path-integral representation of the partition function, a weighted summation of all possible configurations where the trajectories of particles are closed loops. These configurations form the $Z$ configuration space. The worm algorithm works in an enlarged $G$ configuration space by introducing an open-ended worldline called a "worm". The worm's "head" and "tail" correspond to $b$ and $b^\dagger$ operators, respectively. Conventionally, the $b$-point is called *ira*, and the $b^\dagger$-point is called *masha*. Through local updates of *ira* and *masha*, the algorithm efficiently samples the configuration of the partition function and the Green's function of the model. Although the worm algorithm uses a local update scheme, it generally has a smaller dynamical critical exponent than the Metropolis-type updates; thus, it can be more efficient near a phase transition. It is a versatile algorithm that can be applied to various models, including the extended Bose-Hubbard model [29].

In this work, we integrated the clock sampling technique with the worm algorithm and developed the *clock factorized worm algorithm* to simulate EBHM. The algorithm adopts the standard path-integral representation of EBHM, where the basis of Fock states is used as the computational basis. The Fock states are defined as the set of all occupation numbers on each lattice site, $|n_1, n_2, \ldots, n_N\rangle$, where the occupation number $n_i$ on the $i$-th site can take any positive integer value ranging from 0 to $\infty$. The trajectories of the bosons form closed loops in the configuration, and the points in imaginary time where the system changes occupation number are called kinks. We adopted a standard worm update scheme for EBHM consisting of four types of updates: (a) create/delete worm, (b) move worm head, (c) insert/delete kink before the worm head, (d) insert/delete kink after the worm head [80]. The first pair of operations creates a worm or deletes the worm, switching configuration between the $Z$ space and $G$ space. The move worm head operation works in the $G$ space. It shifts one worm head in the imaginary time direction. The insert/delete kink operation inserts/deletes one kink before or after the worm head and changes the spatial position of the worm head. The worm creation is the only possible update when the system is in $Z$ space, while in the $G$ space, updates are chosen randomly according to an *a priori* probability distribution. The detailed description of the worm update scheme can be found in Ref. [80].

Similar to the clock factorized Metropolis algorithm, these updates are local updates, and we use the clock sampling process to handle the long-range interaction terms. The factorized Metropolis filters of all these updates have the standard form $P_{fac} = P_\mathcal{A} \prod_j P_j$, where $P_\mathcal{A}$ depends on specific details of the update and $P_j$ is universal for all types of updates. Updates (a) and (b) change the occupation number within a segment on a single site $i$. Since the long-range interaction strength $V$ is positive in this model, only updates that increase the occupation number are relevant in the factorized Metropolis filter. On the other hand, in updates (c) and (d), the worm head jumps to another site, thus changing the segments on both the starting site and the destination. Although kink operations change two segments simultaneously, the

factorized Metropolis filter can have the same form as updates (a) and (b). This is because, after a kink operation, the occupation number of one segment increases while the occupation number of the other segment decreases. The long-range interactions between the segment with decreasing occupation number and the segment on other sites always lead to an energy decrease, regardless of the configuration; thus, their corresponding factors will not affect the sampling process with $P_j = 1$. Therefore, only the interaction terms related to the segment with the increased occupation number should be considered in the factorized Metropolis filter.

Here, as a simple illustration, we present the $P_\mathcal{A}$ for creating worm update and inserting kink before worm head.

*Create worm.* To create a worm, one randomly selects an existing segment on the $i$-th worldline. The selected segment spans from $\tau_{\min}$ to $\tau_{\max}$ and has an occupation of $n$. Then, one uniformly draws two points $\tau_1, \tau_2$ within the segment as the positions for inserting *ira* and *masha*. The worm deletion is the reverse process of worm creation, which is only possible when *ira* and *masha* are on the same worldline, and there are no kinks between them. Therefore the $P_\mathcal{A}$ for worm creation update is given by,

$$P_\mathcal{A}^{\text{crea}} = \min\left\{1, N_{\text{seg}}\omega_G p_{\text{del}}(\tau_{\max} - \tau_{\min})^2 \times K(\mathcal{S} \to \mathcal{S}')\exp[-\Delta U_{\text{loc}}]\right\}, \tag{31}$$

where $N_{\text{seg}}$ is the number of segments in the configuration, $\omega_G$ is a free parameter to control the relative weight between $Z$ space and $G$ space, $p_{\text{del}}$ is the probability of choosing the delete worm update. The $K(\mathcal{S} \to \mathcal{S}')$ is the off-diagonal weight ratio due to *ira* and *masha* and $\Delta U_{\text{loc}}$ is the local energy difference caused by on-site repulsion and chemical potential.

*Insert kink before ira.* Assuming *ira* is on the $i$-th worldline, we select one of its neighboring worldline $j$ and identify the first kink on the $j$ that is before *ira*, with $\tau_{\min} < \tau_{ira}$. One randomly select a point $\tau_k$ between $\tau_{\min}$ and $\tau_{ira}$, and inserts a new kink $c_i c_j^\dagger$ at $\tau_k$. *Ira* is then shifted to the $j$-th worldline. The $P_\mathcal{A}$ of this update is then given by,

$$P_\mathcal{A} = \min\left\{1, N_{nn} t_{ij}(\tau_{ira} - \tau_{\min}) \times K(\mathcal{S} \to \mathcal{S}')\exp[-\Delta U_{\text{loc}}]\right\}. \tag{32}$$

Here, $N_{nn}$ is the number of nearest neighbors, and $t_{ij} \equiv t$ is the hopping strength between worldline $i$ and $j$. The $K(\mathcal{S} \to \mathcal{S}')$ is the off-diagonal weight ratio due to the insertion of kink and spatial-shift of *ira*, while $\Delta U_{\text{loc}}$ is the local energy difference caused by on-site repulsion and chemical potential.

As stated above, while the $P_\mathcal{A}$ depends on the update, $P_j$ has a general form. Consider a general transition that increases the occupation between $\tau_1$ and $\tau_2$ on the $i$-th site, the factors $P_j$ has the form,

$$P_j = \exp\left\{-\left[V_{ij}\Delta n_i \int_{\tau_1}^{\tau_2} n_j(\tau)\,d\tau\right]^+\right\}. \tag{33}$$

Here, $V_{ij}$ is the interaction strength between spins $i$ and $j$ given by $V_{ij} = V/r_{ij}^\alpha$ and $\Delta n_i = +1$. The bound hazard rate is then given by,

$$\hat{h}_j = 1 - \exp\left\{-V_{ij}n_{\max}|\tau_2 - \tau_1|\right\}. \tag{34}$$

This corresponds to the situation that the segment on the $j$-th site is maximally occupied, where $n_{max}$ is the largest segment occupation in the current configuration. In theory, an arbitrary number of bosons can occupy one site; thus, the value of $n_{\max}$ is not bounded. In practice, one can impose an upper limit on the occupation number of a segment as long as this upper limit covers the Hilbert space being studied. This allows one to determine the bound hazard rate and perform clock sampling. However, using a constant $n_{\max}$ will decrease the algorithm's performance. In this implementation, we use a histogram to keep tracking the maximal occupation number of the current configuration. At the beginning of the simulation,

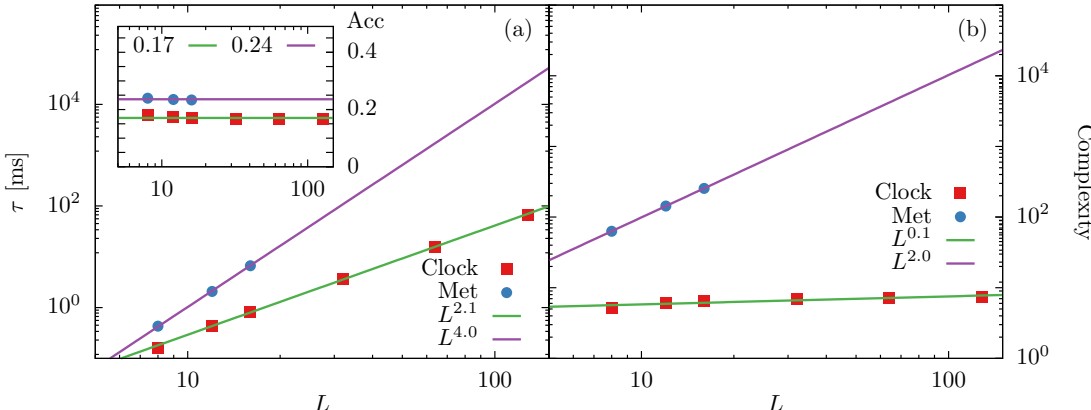

Figure 6: Performance benchmark of the clock factorized worm algorithm compared with the conventional worm algorithm for 2D EBHM at $\alpha = 3.0$, $U/t = 10$, $V/t = 7$, $\mu/t = 0$ and $\beta = 10$. Panel (a) plots the average time per sweep $\tau$ in milliseconds for both algorithms, with an inset presenting the average acceptance ratio. Panel (b) displays the computational complexity per update for both cases.

a histogram is created to record the frequency distribution of the segment occupation number and keep it updated during the simulation. When a new segment is added to the configuration, the histogram records its occupation number, while if a segment with occupation $n_i$ is removed, the corresponding bin in the histogram decreases by one. Therefore, one can keep track of the actual largest segment occupation of the current configuration and ensure the best performance of the clock sampling.

Notably, applying the factorized Metropolis filter does not affect the measurement of Green's function in the worm algorithm. Green's function can be obtained by recording a histogram of the space-time separations between the worm's head and tail and then normalizing this histogram using the ratio between $Z_G$ and $Z$. Since the clock factorized QMC method does not modify the configuration weights in either the G-space or the Z-space, the measurement of Green's function remains unaffected.

Simulations are conducted using the clock factorized worm algorithm to test the efficiency and robustness of the algorithm for the extended Bose-Hubbard Model. Various exponents of the long-range interaction and system sizes are explored, and the computational complexities are compared. The results are shown in Fig. 3(a) and Fig. 4(a) for 2D square and 3D cubic lattices, respectively. The simulations are conducted $U/t = 10$, $\mu/t = 0$, and $V/t = 7$ with the inverse temperature fixed at $\beta = 10$. The observed computational complexity for different system sizes increases much slower than $L^d$, demonstrating a significant improvement in the simulation efficiency of the clock factorized worm algorithm. Fig. 6 illustrates the performance benchmarks on the EBHM on a 2D square lattice with $\alpha = 3.0$, $U/t = 10$, $V/t = 7$, $\mu/t = 0$ and $\beta = 10$. Similar to the LRTFIM case, the result demonstrates a near $\mathcal{O}(N)$ reduction of time per sweep $\tau$ for the clock factorized worm algorithm, approximately in the same order as the reduction of the computational complexity. Moreover, the acceptance ratios of both algorithms are almost independent of system sizes, dropping from $P_{\text{acc}} \approx 0.23$ to $0.18$. Hence, for EBHM, the overall efficiency improvement of the new algorithm is $\mathcal{O}(N)$.

## 4.3 Clock factorized worm algorithm with long-range hopping

The long-range XXZ Heisenberg Model is a theoretical model used in condensed matter physics to describe the behavior of interacting spins in a lattice structure. The model is an extension of the XXZ Heisenberg model, which includes both nearest-neighbor and next-nearest-neighbor

spin interactions. In the long-range XXZ Heisenberg model, the spin interactions can be long-range and exhibit power-law decay with distance. This model has been widely studied in both theoretical [23, 81–83] and experimental contexts [84] due to its relevance in describing the properties of spin systems in a variety of physical systems, including magnetism, superconductivity, and quantum computing. The long-range XXZ Heisenberg Model has proven to be a valuable tool for understanding the complex behavior of interacting spin systems in lattice structures and has led to important insights into the nature of quantum phase transitions and critical phenomena.

The Hamiltonian of the LRXXZ model is given by,

$$\mathcal{H} = -\sum_{i<j} \frac{1}{r_{ij}^\alpha} \left[ J^x \left( S_i^x S_j^x + S_i^y S_j^y \right) - J^z S_i^z S_j^z \right],$$ (35)

where $S_i^\beta (\beta = x, y, z)$ is the quantum-spin operators attached to each site. $J^x$ is in-plane ferromagnetic interactions leading to a sign-positive model, while $J^z$ is the amplitude for $S_i^z S_j^z$ interactions. The LRXZZ model can be mapped to a hard-core boson model by using the transformation $S_i^x + iS_i^y = b_i^\dagger$ and $S_i^z = n_i - 1/2$. The Hamiltonian describes the mapped model,

$$\mathcal{H} = -t \sum_{i<j} \frac{1}{r_{ij}^\alpha} \left( b_i^\dagger b_j + h.c. \right) + V \sum_{i<j} \frac{1}{r_{ij}^\alpha} n_i n_j - \sum_i \mu n_i,$$ (36)

where $t = -J^x/2$, $V = J^z$ and $\mu = J^z/2 \sum_{j>0} 1/r_{0j}^\alpha$. A constant term is dropped after the mapping. For the hard-core boson model, the occupation number is restricted to only 0 and 1. The hard-core boson model can also be simulated using the clock factorized worm algorithm by setting a hard limit on the max occupation number. Any updates that result in a segment with an occupation number larger than 1 are rejected.

The update scheme and clock sampling process are identical to the previous algorithm, except that now we allow additional long-range hopping terms, i.e., the destination of kink operation is not limited to nearest-neighboring sites. For example, consider a spatial shift of *ira* by inserting a new kink before ira. For long-range hopping cases, the destination $j$ of the hopping can be selected from all the rest of the worldlines according to a probability distribution $\mathcal{A}(i \rightarrow j)$. The $P_\mathcal{A}$ of this update is similar to Eq. (32):

$$P_\mathcal{A} = \min\left\{ 1, \frac{t_{ij}}{\mathcal{A}(i \rightarrow j)} \left( \tau_{ira} - \tau_{\min} \right) \times K(\mathcal{S} \rightarrow \mathcal{S}') \exp\left[ -\Delta U_{\text{loc}} \right] \right\}.$$ (37)

Suppose the hopping destination is uniformly chosen from all possible sites, i.e., $\mathcal{A}(i \rightarrow j) = 1/(N-1)$. For long-range hopping strength with the form $t_{ij} = t/r_{ij}^\alpha$ with $r_{ij}$ being the distance between site $i$ and site $j$, the acceptance probability of a kink-insertion update will also decay algebraically with the distance of hopping. In that case, the long-range hopping update will hardly be accepted, significantly hindering the algorithm's efficiency.

Our solution to this problem is to propose the hopping destinations $j$ according to a probability distribution of the distance of the hopping,

$$\mathcal{A}(i \rightarrow j) = c \frac{t}{r_{ij}^\alpha},$$ (38)

where $c$ is a normalization constant such that,

$$c \sum_{j \neq i} \frac{t}{r_{ij}^\alpha} = 1,$$ (39)

where the sum goes over all possible neighbors. The probability of proposing hopping with longer displacement is algebraically suppressed. This distribution $\mathcal{A}(i \rightarrow j)$ can cancel the $t_{ij}$

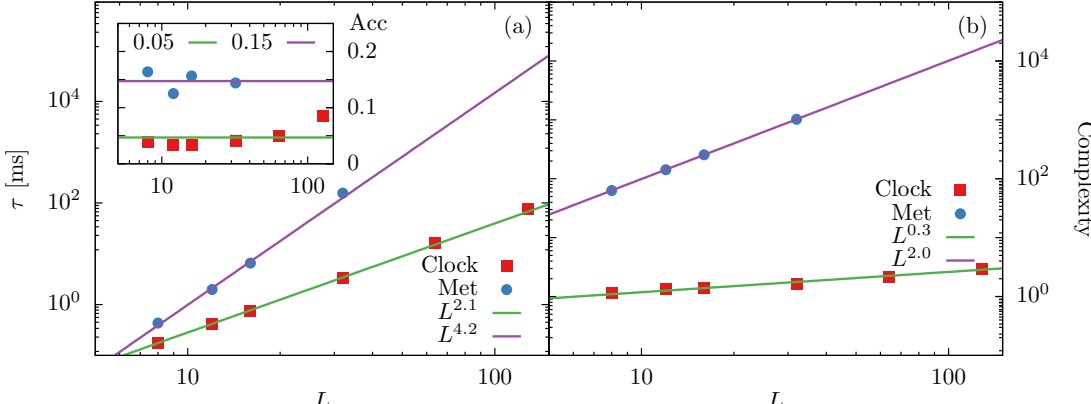

Figure 7: Performance benchmark of the clock factorized QMC algorithm compared with the conventional QMC algorithm for 2D LRXXZ model at $\alpha = 3.0$ and $J_z/J_x = 9$. Panel (a) shows the average time per sweep $\tau$ in milliseconds for both algorithms, with an inset presenting the average acceptance ratio. Panel (b) displays the computational complexity per update for both cases.

term in the expression of $P_{\mathcal{A}}$ up to a constant $c$; thus, this distribution increases the overall acceptance ratio of long-range hopping updates in the worm algorithm. Since $\mathcal{A}(i \rightarrow j)$ only depends on the lattice and the long-range hopping, one can compute all the elements of the distribution before the simulation and sample it using Walker's alias method, as described in A.2. With this technique, the algorithm can efficiently handle diagonal and off-diagonal long-range interactions.

Simulations are conducted for the long-range XXZ Model to test the efficiency of the clock factorized worm algorithm with long-range hopping. Various exponents of the long-range interaction and system sizes are explored, and the computational complexities are compared. The results are shown in Fig. 3(a) and Fig. 4(a) for 2D square and 3D cubic lattices, respectively. The simulations are conducted with $J^x/J^z = 1$ and the inverse temperature fixed at $\beta = 10$. The observed computational complexity for different system sizes increases much slower than $L^d$, demonstrating a significant improvement in simulation efficiency. Fig. 7 shows the performance benchmarks on the LRXXZ on a 2D square lattice with $\alpha = 3.0$, $J_z/J_x = 9$, and $\beta = 10$ [85]. Similar to previous cases, the result demonstrates a near $\mathcal{O}(N)$ reduction of time per sweep $\tau$ for the clock factorized worm algorithm with long-ranged hopping. The average acceptance ratio of the new algorithm drops from $P_{\text{Acc}} \approx 0.15$ to $0.05$, roughly by a constant ratio. Despite the constant increase in autocorrelation time due to this drop, the computational complexity per update of the new algorithm remains $\mathcal{O}(1)$; thus, overall performance improvement in this case should also scale as $\mathcal{O}(N)$.

Note that the low acceptance rate of the clock factorize QMC algorithm is parameter-dependent and can be addressed through targeted optimizations, such as the box technique mentioned in the paper, which helps compensate for a decrease in acceptance rate. In this specific case, the long-range XXZ model is mapped to a hard-core Bose-Hubbard model with long-range hopping terms. For $J_z/J = 9$, this mapping introduces a local chemical potential term $\mu \sim 34$. The rejection factor corresponding to this local field is then given by

$$\int \exp(\mu \Delta n_i) d\tau, \tag{40}$$

which may heavily suppress the acceptance ratio of updates. To mitigate this, one can apply the box technique by incorporating interaction between nearest neighbors into this factor:

$$\int \exp\left( \mu \Delta n_i - \sum_{\langle i,j \rangle} \frac{V}{r_{ij}^{\alpha}} \Delta n_i n_j \right) d\tau \,, \tag{41}$$

where $\sum_{\langle i,j \rangle}$ sums over nearest-neighbors of $i$. This trick can improve the acceptance rate while retaining the $\mathcal{O}(1)$ computational complexity. Additionally, other strategies, such as sampling the worm move distance or the head-tail separation from an exponentially decaying distribution, can further improve the efficiency of the worm algorithm [86, 87].

## 5 Discussion and outlook

In summary, we develop the clock factorized quantum Monte Carlo method, which is both efficient and generic for simulating long-range interacting quantum systems. We formulate three efficient clock factorized Monte Carlo algorithms with various update schemes tailored specifically for the LRTFIM, EBHM, and LRXXZ. Extensive benchmarks show that, compared with the conventional Metropolis algorithms, there is a significant efficiency improvement for these novel algorithms. For non-frustrated systems, incorporating bound rejection and introducing first-bound-rejection events on a tree structure can lead to significant acceleration with computational complexity scaling as $\mathtt{A} \sim \mathcal{O}(N)$ for strictly extensive systems, $\mathtt{A} \sim \mathcal{O}(N^{\kappa})\,(0 < \kappa < 1)$ for sub-extensive systems, and $\mathcal{O}(N/(\ln N^2)) < (\mathtt{A})_{\text{margin}} < \mathcal{O}(N/\ln(N))$ for marginally extensive systems. Analysis and optimization of the algorithm performance for the sub-extensive long-range quantum system could be an interesting future direction. For systems with frustrated long-range interaction in their diagonal terms, combining the clock factorized QMC technique with the box technique effectively reduces computational complexity while incurring only a slight decrease in the acceptance ratio [34].

Our method is not only efficient but also flexible and independent of the update scheme. Besides the local update and worm updates, the recursive clock sampling technique can be applied to the cluster Monte Carlo method because the bond activation events are intrinsically independent of each other. Notably, its recent application to the 2D classical O($n$) spin model with long-range coupling [88, 89] demonstrates both the efficiency and versatility of the algorithm. The extended cluster algorithms for long-range interacting spin systems [37, 90] can be understood as specific cases of the recursive clock sampling method. Moreover, the recursive clock sampling method is a more general technique than the Metropolis method, with the latter being a limiting case of the former. This implies that the clock factorized QMC method is at least as effective as the Metropolis method in terms of performance.

In the current study, we perform benchmarks at fixed temperatures to demonstrate the power of the clock technique in reducing the computational complexity due to long-range interactions. In path-integral QMC simulations, to extract the quantum critical properties, one typically performs simulations at an inverse temperature $\beta$ that scales as the system size $L$, for instance, $\beta \approx L^z$, where $z$ is the dynamical critical exponent of the quantum phase transition. The computational effort for such simulations generally scales linearly with $\beta$ due to the strictly short-range nature of interactions in the imaginary time dimension. The clock technique, specifically designed to address the complexity of long-range interactions in the spatial dimension, dramatically improves overall scaling performance. Thus, the overall scaling of the computational effort per sweep for a long-range interacting system of size $N$ at inverse temperature $\beta$ is: $O(\beta N^2)$ for conventional algorithms $O(\beta N \times C)$ for the clock QMC method.

It is worth clarifying that the efficiency of measuring observables is generally independent of update strategies in QMC simulations. Therefore, improving update efficiency is always worth the effort. Due to critical slowing down near the critical point, the autocorrelation between consecutive measurements is generally large. To generate effectively independent samples, one needs to perform multiple sweeps between two measurements. Therefore, improving the efficiency of updates in long-range interacting systems is crucial. While measuring certain quantities in the long-range interacting system has an $\mathcal{O}(N^2)$ algorithmic complexity, regardless of the update strategies, not all required physical quantities necessitate this level of measurement. For example, in the study of magnetism, the order parameter and related quantities are only at an $\mathcal{O}(N)$ level, thus do not introduce additional computational cost. Moreover, introducing long-range interactions does not necessarily increase the measurement complexity for many observables, such as the correlation functions, which can still be measured within a $\mathcal{O}(N)$ algorithmic complexity. Furthermore, certain update schemes allow for the definition of improved estimators, enabling the efficient measurement of these quantities. In cases where energy-like physical quantities are essential, one can measure quantities such as nearest-neighbor energy and specific heat, which should exhibit similar scaling behavior near the critical points. For comprehensive energy measurements, optimized methods utilizing fast Fourier transformation (FFT) can be employed to reduce the computational cost [91]. These methods are generally independent of update strategies, emphasizing the importance of developing efficient algorithms for Monte Carlo updates.

We also note that, for the LRTFIM, stochastic series expansion methods have been successfully used to achieve efficient simulations with an overall complexity of $O(N \log N)$ [11,92]. While SSE is a well-established approach for studying quantum spin systems [24,93] and can be applied to quantum systems with long-range interactions (see Ref [92] for detailed discussion), our work takes an alternative route by developing an efficient algorithm within the PIMC framework. In fact, this study marks the first step toward efficient PIMC algorithms for long-range systems. Our method is designed to handle long-range interactions and incorporate off-diagonal contributions within the path-integral formalism, making it applicable to a broad range of quantum systems, such as the extended Bose-Hubbard model. The two approaches address different challenges and offer complementary advantages [92], and a detailed comparison between SSE and PIMC is beyond the scope of this work and remains an interesting direction for future research.

Considering the recent active studies on long-range interacting systems that heavily rely on Monte Carlo simulations and recent focus on the development of efficient classical Monte Carlo methods [94], the clock factorized QMC method, due to its simplicity and ease of use, can provide a readily available tool to explore the rich physics of these systems and is a promising candidate for studying long-range interacting systems in various fields of physics. The Rydberg atom array is a crucial platform for studying quantum computation [95] and exploring exotic phases like quantum spin liquids [96]. Despite recent advancements in both theory and experiments [96–99], numerical simulations of these systems remain challenging due to long-range interactions [60,100]. The clock factorized QMC method enables large-scale simulations of Rydberg atom arrays without truncating van der Waals interactions or other approximations, allowing for an unbiased investigation of the system. This could offer valuable insights and guide future theoretical and experimental developments of the Rydberg system [101]. Another potential application is to combine the recursive clock sampling technique with the worm algorithm of the continuous-space path-integral Monte Carlo method [10], the state-of-the-art method for studying long-range interacting bosonic gases, such as dipolar bosonic gas system, which is closely related to AMO experiments. The worm head update in this algorithm can be sampled using the recursive clock sampling technique to efficiently account for the long-range interactions, allowing for simulations of the system with larger particle numbers.

## Acknowledgments

**Funding information**   This work has been supported by the National Natural Science Foundation of China (under Grant No. 12275263 and 12204173), the Innovation Program for Quantum Science and Technology (under Grant No. 2021ZD0301900), and the Natural Science Foundation of Fujian Province of China (under Grant No. 2023J02032).

## A   Related algorithm

### A.1   Inversion method

Inverse transform sampling, or inversion method, is one of the most simple and universal techniques for generating random numbers from a discrete probability distribution given its cumulative distribution function. For a discrete random variable $X$ with PMF $p(k)$, the cumulative distribution function (CDF) quantifies the likelihood that a random variable does not exceed the $k$: $F(k) = \sum_{i=1}^{k} p(i)$. The inversion method generates the random number $X$ via the corresponding inverse of CDF:

$$X = F^{-1}(u) = \min\{k : F(k) \geq u\}, \tag{A.1}$$

where $u \equiv \mathtt{ran}$ is a uniform random variable and min is the minimum function that returns the smallest $k$ that satisfies the condition. Hence, once the inverse CDF of the target distribution is known, one can generate $X$ using one uniform random number. However, obtaining a simple closed form of $F^{-1}(u)$ is difficult except for a few classes of discrete distributions. One of the most useful discrete distributions that can be easily generated via inverse CDF is the geometric distribution, which is also relevant to clock sampling.

Consider a long-range interaction model on a complete graph, where every site interacts with all other sites with identical strength $J$. One can define a constant bound hazard rate $\hat{h}$ for all factors; thus, the distribution of the bound rejection events follows a geometric distribution $P(X = k) = p(1-p)^{k-1}$, with parameter $p \equiv \hat{h}$. The CDF of the geometric distribution is $F(k) = 1-(1-p)^k$. The inverse CDF function is then given by,

$$\begin{aligned} F^{-1}(u) &= \min\{k : 1-(1-p)^k \geq u\} \\ &= \min\{k : k \geq \log(1-u)/\log(1-p)\} \\ &= \left\lceil \frac{\log(u)}{\log(1-p)} \right\rceil, \end{aligned} \tag{A.2}$$

where $\lceil x \rceil$ is the ceiling function that returns the smallest integer larger than or equal to $x$. Therefore, the random variable $X = F^{-1}(\mathtt{ran})$ is geometrically distributed.

This method is particularly important because, at each level of clock sampling, geometric distribution can be used to sample the bound rejection events by setting a constant bound hazard rate $\hat{h}$ for all $P_j$ factors of the current tree level. The original clock technique for long-range interacting classical systems can be viewed as a clock sampling process using geometric random numbers to sample bound rejection events at each level of the tree [34].

Although the analytical form of $F^{-1}(u)$ is generally inaccessible for an arbitrary discrete distribution, the inversion method allows one to evaluate $F^{-1}(u)$ by solving the inversion inequality:

$$F(X-1) < u \leq F(X). \tag{A.3}$$

Generating a random variable using the inverse CDF is equivalent to solving $X$ for the above inequality, with $u$ being a uniform random number. An exact solution of the inversion inequality always exists and can be found in finite time [54]. This property of the inversion

method makes it universally applicable for generating random numbers from a wide range of distributions, even if their inverse CDF cannot be expressed in a closed analytical form.

Various algorithms exist to solve the inversion inequality. One of the simplest methods is the *sequential search*, where the solution of inversion inequality is searched sequentially starting from 0. In this method, one generates a uniform random number $u$ and evaluates the CDF function on the fly until the first $k$ value satisfies $F(k) >= u$. The expected number of iterations is $E(X) + 1$, where $E(X)$ is the expectation of random number $X$. Thus, the performance of the sequential search algorithm depends on the tail of the target distribution $p(k)$. The performance of the sequential search algorithm can be improved using several techniques, such as a binary search or a table-aided search method [54, 57]. However, these algorithms usually have a slow setup process and, therefore, are not optimal for generating bound rejections whose distribution varies during the simulation.

## A.2 Walker's alias method

Besides the inversion method, another commonly employed algorithm for efficient sampling from discrete probability distributions is Walker's alias method, which was originally devised by A. J. Walker in 1974 [55, 56]. Like the inversion method through sequential search, the alias method requires a slow setup, rendering it suboptimal for generating the bound rejection events. Nevertheless, we include it for the sake of completeness, and more importantly, it proves to be valuable when handling long-range off-diagonal interactions, as will be discussed in section 4.

---

**Algorithm 4:** Alias table setup

**Input:** Discrete probability $p_k$, $k \in 0, 1, 2, \ldots, N$
**Output:** Alias array $a(k)$ and probability array $q(k)$
**for** $k = 1, 2, \ldots, N$ **do**
$\quad$ | $\quad q(k) \leftarrow N * p(k)$ and $a(k) \leftarrow k$;
**end**
Initialize $Rich = \{q(k) \geq 1\}$ and $Poor = \{q(k) < 1\}$;
**while** *Poor* and *Rich* are not empty **do**
$\quad$ | $\quad$ Randomly pick $\ell \in Poor$ and $\hbar \in Rich$;
$\quad$ | $\quad$ Set alias $a(\ell) \leftarrow \hbar$;
$\quad$ | $\quad$ Remove element $\ell$ from the *Poor* array;
$\quad$ | $\quad$ Set $q(\hbar) \leftarrow q(\hbar) - (1 - q(\ell))$;
$\quad$ | $\quad$ **if** $q(\hbar) < 1$ **then**
$\quad$ | $\quad$ | $\quad$ Move $\hbar$ from *Rich* to *Poor*.
$\quad$ | $\quad$ **end**
**end**
**for** any remaining element $k$ in *Poor* or *Rich* **do**
$\quad$ | $\quad$ Set $q(k) \leftarrow 1$
**end**

---

Given a discrete probability distribution $p(k)$ with $k \in \{1, 2, \ldots, N\}$, let probabilities be amplified by a factor of $N$ so that the averaged probability is now 1, instead of $1/N$. Then, one splits the elements of the probability distribution into three classes: for each element $k$, label it as "poor" if $p_k < 1$, as "rich" if $p_k > 1$, or as "average" if $p_k = 1$. The basic idea of setting up Walker's alias method is the "Robin Hood Rule": taking from the "rich" to bring the "poor" up to average [102]. Specifically, one takes the probability of a "rich" element, $\hbar$, and gives it to some "poor" element, say $\ell$ to bring it up to the averaged value 1, i.e., the amount

Target Distribution

| $k$ | 1 | 2 | 3 | 4 |
|---|---|---|---|---|
| $p(k)$ | 1/2 | 1/3 | 1/12 | 1/12 |

Alias Table

| $k$ | 1 | 2 | 3 | 4 |
|---|---|---|---|---|
| $q(k)$ | 2/3 | 1 | 1/3 | 1/3 |
| $a(k)$ | 2 | 2 | 1 | 1 |

Figure 8: An example of the alias method for a discrete distribution of 4 elements. For a target distribution $p(k)$, a possible alias table is shown.

of probability taken is $\delta_\ell = 1 - p_\ell$. For the "poor" to record its donor, its corresponding alias index is set to $a_\ell \leftarrow \hbar$. In addition, the remaining probability of element $\hbar$ is recorded as $q(\hbar) \leftarrow q(\hbar) - \delta_\ell$. After the donation, the "poor" element $\ell$ is labeled as "average," while the "rich" element, with a remaining amount $p_\hbar - \delta_\ell$, might become below the average and, if so, it is re-labeled as "poor". This process is repeated until no "rich" or "poor" element is left. If either the "rich" or "poor" category empties before the other, $q(k)$ of the remaining entries are set to 1 with negligible error [103]. Notice that in each step, the size of "average" elements increases at least by one; thus, the setup process has a time complexity of $\mathcal{O}(n)$. The pseudocode code for setting up the alias table is described in Alg. 4.

After building up the alias table, one can easily sample the target distribution $p(k)$ in two steps: firstly, one uniformly draws an entry $i$ from the alias table. Then one generate an uniform random number `ran`, if `ran` $< q(i)$, return $i$; otherwise, return its alias $a(i)$. The resulting random number conforms to the target distribution $p(k)$. Sampling a discrete distribution via the alias method has a time complexity of $\mathcal{O}(1)$ because it only involves a single comparison and less than two table accesses.

In conclusion, Walker's alias method provides an efficient algorithm for sampling from discrete probability distributions. By employing an alias table, random numbers can be generated with $\mathcal{O}(1)$ time complexity. The setup of the alias table can be accomplished using the Robin Hood Rule, redistributing probabilities from "rich" to "poor" elements. Overall, Walker's alias method offers a valuable approach for efficient sampling and has been widely used in Monte Carlo simulations and other probabilistic algorithms.

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
