# Peer review of "Clock Factorized Quantum Monte Carlo Method for Long-range Interacting Systems"

_SciPost Physics Core, doi:SciPost Phys. Core 8, 036 (2025)_

## Round 1 · Referee Report · Anonymous (Referee 1) · 2024-7-24

Strengths

1) Novel quantum Monte Carlo approach to simulate quantum systems with long-range interactions 2) Flexible method which can be incorporated into existing codes 3) Detailed self-contained presentation and discussion of the method

Weaknesses

1) Unclear what is new in the first 3 sections 2) No physics shown

Report

In the manuscript "Clock Factorized Quantum Monte Carlo Method for Long-range Interacting Systems", the authors present a path integral quantum Monte Carlo (PIMC) approach to simulate quantum systems with long-range interactions. This is done by generalizing a previously introduced method for classical systems in [Phys. Rev. E 99 010105 (2019)]. The authors provide a detailed, clear, and self-contained discussion of their method. In particular, they further show the implementation for several quantum models, which comes with a generalization of the worm algorithm for long-range interactions.

Developing efficient and unbiased quantum Monte Carlo (QMC) methods for quantum systems with long-range interactions is of importance, and this manuscript provides a good contribution to this field. Further, the "clock-factorized method" can be incorporated into existing QMC codes. In my opinion, this is clearly interesting work and does meet the SciPost criterion "Open a new pathway in an existing or a new research direction, with clear potential for multi-pronged follow-up work".

Before recommending publishing it, there are, however, a few points I kindly ask the authors to address, which are listed in 1)-3) below.

Requested changes

1) In the abstract the authors mention that in addition to the PIMC generalization, this manuscript contains further technical improvements. To me, it is, however, not really clear what is new and what isn't in the first three sections. While I highly appreciate the self-contained nature of the discussion, it should be made more transparent in case this is largely a review of the approach presented in [Phys. Rev. E 99 010105 (2019)]. If there are some technical improvements, highlighting them more could also further strengthen the manuscript.

2) In terms of efficiency and complexity for the transverse field Ising model (TFIM) they compare their method to a "naive" PIMC implementation with complexity $\mathcal{O}(N^2)$. For the TFIM there already exist, however, efficient QMC methods with overall complexity $\mathcal{O}(N\log(N))$, which to my knowledge can also be extended to the XXZ model (see DOI 10.1103/PhysRevE.68.056701 and related works). In my opinion, this is a more relevant comparison for their method that should be discussed.

3) To my understanding of the method, there is essentially a tradeoff between a lower overall acceptance rate but (much) less computation time per Monte Carlo step. Further the authors argue, that the clock factorized algorithm does not change the dynamics of the algorithm and resort to only showing the complexity/runtime per update step. I think, however, it would still be beneficial to show the efficiency in terms of measuring some actual physical observable, either in terms of computation time to reach similar accuracy (for a comparison) or showing results for system sizes previously not reachable. Since the acceptance rate for the XXZ model algorithm is as low as around 5% this could further erase additional doubts about the efficiency.

Following are a few minor/optional questions and comments that the authors could consider for their revision.

4) The authors state that the overall "complexity" per step is given by $C\sim \mathcal{O}(\log(P_B)/\log(P_{\mathrm{fac}}))$, which to my understanding is the key part for the efficiency since in case where $P_B$ scales with $N$ the same way as $P_\mathrm{fac}$ a complexity of $\mathcal{O}(1)$ is achieved. Since $P_B$ depends on the choice of the hazard rates, my question is: When can the condition that $P_B$ scales with $N$ like $P_\mathrm{fac}$ be met? Is this always the case for extensive systems? A clarification in the manuscript might be beneficial and could provide more clarity for the QMC community when to use the method.

5) While it was clearly introduced, the term "complexity" used in the manuscript could still be a bit misleading as it only refers to the complexity per update step (of which $N$ are performed). Changing it to something like "complexity per step" (also in the figures) could thus be more appropriate.

6) The authors mentioned that the clock factorized method does not affect the general dynamics of the worm algorithm. Does the clock factorized approach affect the measurement of Green's functions? It might be beneficial to (shortly) comment on this in the manuscript.

7) I have to admit being a bit confused regarding the statements for frustrated systems in the discussion. Can the box technique here help reduce the QMC sign problem?

While the article is overall very well-written, here are some typos that caught my eye, which the authors may want to address. In the following "l." refers to line.

  • several times "trails" is used instead of "trials"
  • l.94: which "can" be considered as a generalization
  • l. 377: efficently → efficiently
  • l.509 Is this one sentence instead of two? (, one starts with)
  • l.519 hardly occur (without s)
  • l.789 allows one "to" determine

In several instances some article seems to be missing: - l.192 "an" a priori - l.391 "a" configuration independent - l.706 "the" EBHM

Recommendation

Ask for minor revision

---

## Round 1 · Referee Report · Anonymous (Referee 2) · 2024-9-8

Strengths

1) Novel quantum Monte Carlo algorithm for quantum systems with long-range interaction with potentially improved performance.

Weaknesses

1) Calculated quantity not well defined. 2) No physical results calculated. 3) Work and results are not well embedded in existing literature.

Report

The article entitled "Clock Factorized Quantum Monte Carlo Method for Long-range Interacting Systems" by Fan, Zhang, and Deng describes an improved quantum Monte Carlo algorithm in the path-integral formalism to treat quantum systems with long-range interactions. This novel algorithm is called clock factorized quantum Monte Carlo and is adapted from classical to quantum systems by the authors. The latter are intensively studied in current research and are relevant for quantum-optical platforms and condensed matter systems. On the one hand, the described method sounds interesting (and flexible) and the article is well written. On the other hand, the algorithm is not well embedded in the current status of research and literature and the presented results are of very limited use. In fact, the present article does not contain any new physical result. In case the authors are able to resolve the issues mentioned, the article is in my opinion therefore better suited for SciPost Core.

Requested changes

Specifically, let me address the following major points:

1) In chapter 4, for all examples, the authors present numerical results of the new algorithm for a quantity which is called "Complexity". I think it would be important to define this quantity precisely in an equation and refer to this equation and quantity whenever possible.

2) In chapter 4, for all examples, it would be important to also include physical quantities like the ground-state energy or order parameter and compare the obtained results quantitatively with the known ones from literature. Current simulations are done at rather large temperatures. How do the simulations perform by a proper scaling of temperature and length scales of systems when extracting quantum critical properties?

3) I am surprised that the authors do not refer to stochastic series expansion quantum Monte Carlo pioneered in Sandvik, A.W. Stochastic series expansion method for quantum Ising models with arbitrary interactions. Phys. Rev. E 2003, 68, 056701, which is heavily used for quantum systems with long-range interactions. For more bibligraphy please also check the recent review Entropy 2024, 26(5), 401 on Monte-Carlo approaches to long-range interactions in quantum systems, which in particular coveres also the physics of the three examples discussed by the authors.

Minor points:

4) Lines 33-36: There are no references. 5) Line 299: "production form" - "product form" 6) Line 370: "1-p_j,rel)" -> "(1-p_j,rel)" 7) Line 480: Line too long. 8) Line 605: There are no references. 9) Line 619: Line too long. 10) Line 700: The Bose-Hubbard model is more relevant in quantum optics compared to condensed matter physics.

Recommendation

Accept in alternative Journal (see Report)

---

## Round 2 · Referee Report · Anonymous (Referee 1) · 2025-3-4

Report

In their revised manuscript, the authors have satisfactorily addressed my previous remarks, and I therefore recommend the article for publication.

Recommendation

Publish (meets expectations and criteria for this Journal)

---

## Round 2 · Referee Report · Anonymous (Referee 2) · 2025-3-8

Report

In the revised version the authors have addressed all my points in satisfying fashion. I am therefore recommending publication in SciPost Core.

Recommendation

Publish (meets expectations and criteria for this Journal)

---

## Round 2 · Author Response

Warnings issued while processing user-supplied markup:

  • Inconsistency: Markdown and reStructuredText syntaxes are mixed. Markdown will be used.
    Add "#coerce:reST" or "#coerce:plain" as the first line of your text to force reStructuredText or no markup.
    You may also contact the helpdesk if the formatting is incorrect and you are unable to edit your text.

Dear Editor,

Thank you for your detailed feedback and for forwarding the referees' insightful comments, which prompted us to reconsider and improve our manuscript. After reviewing the acceptance criteria for both SciPost Physics and SciPost Physics Core, we agree that SciPost Physics Core is a more suitable venue for our work. Following the referee's comments, we have carefully revised the manuscript to address all the raised points. The changes are highlighted in blue. We believe that the changes will enhance the work's clarity, completeness, and overall quality.

Thank you again for your time, and we look forward to your continued evaluation of our manuscript.

Sincerely, Zhijie Fan, Chao Zhang and Youjin Deng

Reply to Referee 1 We warmly thank the referee for the careful review and constructive suggestions. Below, we address the referee's specific comments.

Comment 1: In the abstract the authors mention that in addition to the PIMC generalization, this manuscript contains further technical improvements. To me, it is, however, not really clear what is new and what isn't in the first three sections. While I highly appreciate the self-contained nature of the discussion, it should be made more transparent in case this is largely a review of the approach presented in [Phys. Rev. E 99 010105 (2019)]. If there are some technical improvements, highlighting them more could also further strengthen the manuscript.

Reply 1: We thank the referee for their insightful comments. In the revised manuscript, we have clarified the relationship between our work and the approach presented in [Phys. Rev. E 99, 010105 (2019)]. While the first three sections provide a self-contained review to ensure readability and accessibility, this manuscript also introduces key technical improvements that extend the original Clock Monte Carlo technique.

In particular, we highlight the following major improvement: the identification and application of a factorized Metropolis filter to the Metropolis-Hastings criterion. This advancement allows the generalization of the Clock Monte Carlo technique to update with asymmetric a-priori probabilities and finite local fields, which is crucial for incorporating off-diagonal terms in Path Integral Monte Carlo (PIMC). This generalization represents a significant step forward, as it enables the method to handle a broader class of quantum systems.

To address the referee's suggestion, we have revised the manuscript to explicitly emphasize these technical improvements and clearly distinguish them from the review components. We believe this will enhance the transparency and impact of our work.

Comment 2: In terms of efficiency and complexity for the transverse field Ising model (TFIM) they compare their method to a "naive" PIMC implementation with complexity. For the TFIM there already exist, however, efficient QMC methods with overall complexity, which to my knowledge can also be extended to the XXZ model (see DOI 10.1103/PhysRevE.68.056701 and related works). In my opinion, this is a more relevant comparison for their method that should be discussed.

Reply 2: We thank the referee for highlighting the Stochastic Series Expansion (SSE) algorithm for long-range interacting systems, as described in [Phys. Rev. E 68, 056701 (2003)]. SSE, with its $O(N\log(N))$ complexity, is an important approach in the study of quantum spin systems. In our manuscript, we focus on a different framework by developing an efficient method within the Path Integral Monte Carlo (PIMC) approach.

The two methods address different challenges and offer complementary advantages. Our algorithm is specifically designed to handle long-range interactions and incorporate off-diagonal contributions within the path-integral formalism, making it applicable to a broad range of quantum systems, such as the extended Bose-Hubbard model.

In the revised manuscript, we have included a brief discussion to provide context on the role of SSE and highlight the distinctions between the two approaches. However, a detailed comparison between SSE and PIMC, particularly for TFIM with long-range interactions, is beyond the scope of this work. Such a comparison would require extensive benchmarking and is an interesting direction for future exploration.

Comment 3: To my understanding of the method, there is essentially a tradeoff between a lower overall acceptance rate but (much) less computation time per Monte Carlo step. Further the authors argue, that the clock factorized algorithm does not change the dynamics of the algorithm and resort to only showing the complexity/runtime per update step. I think, however, it would still be beneficial to show the efficiency in terms of measuring some actual physical observable, either in terms of computation time to reach similar accuracy (for a comparison) or showing results for system sizes previously not reachable. Since the acceptance rate for the XXZ model algorithm is as low as around 5% this could further erase additional doubts about the efficiency.

Reply 3: We thank the referee for raising this important point. The primary focus of this manuscript is to introduce and validate the methodological framework of our Clock Factorized PI-QMC approach, emphasizing its computational efficiency and flexibility. While benchmarking against physical observables is valuable, a comprehensive analysis of specific systems, such as the XXZ model, lies beyond the intended scope of this work.

Regarding the acceptance ratio for the XXZ model, we agree that a lower acceptance rate may raise concerns. However, this limitation is context-dependent and can be addressed through targeted optimizations, such as the box technique mentioned in the paper, which can help compensate for the decrease in acceptance rate. In this particular case, the long-range XXZ model is mapped to a hard-core Bose Hubbard model with long-range hopping terms. For $J_z/J=9$, this mapping introduces a local chemical potential term $\mu \sim 34$. The rejection factor corresponding to this local field is $\int \exp(\mu \Delta n_i)d\tau$, which might heavily suppress the acceptance ratio of updates. To mitigate this, a simple adjustment, such as incorporating interactions between nearest neighbors into this factor, can increase the acceptance rate, $\int \exp(\mu \Delta n_i + \sum_{\langle i,j\rangle}V/r_{ij}^\alpha \Delta n_i n_j)d\tau$. Additionally, other strategies, like sampling the worm move distance or the head-tail separation from an exponentially decaying distribution, could further compensate for the local weight changes. Furthermore, for this specific choice of parameter, the system is deep inside the anti-ferromagnetic phase, where the configuration is mostly fixed with very small fluctuations. These optimizations, however, are model- and parameter-specific, highlighting that any meaningful benchmarking requires a carefully defined context.

A detailed analysis of physical observables, such as computation time to achieve specific accuracies or accessing previously unreachable system sizes, would require significantly more extensive studies. These efforts are currently underway as part of ongoing research projects. In these studies, we aim to systematically investigate quantum critical properties and other physical observables in long-range interacting quantum systems, including Rydberg atom arrays and long-range XXZ models.

In the revised manuscript, we comment on the acceptance ratio for the LRXXZ model and discuss ways to solve the issue. We have also clarified that this work serves as a foundational study, paving the way for future investigations. We hope this addition addresses potential concerns about the algorithm's efficiency while maintaining the manuscript's focus on methodological development.

Comment 4: The authors state that the overall "complexity" per step is given by $C∼O(\log(P_B)/\log(P_{fac})$, which to my understanding is the key part for the efficiency since in case where $P_B$ scales with N the same way as $P_{fac}$ a complexity of $O(1)$ is achieved. Since $P_B$ depends on the choice of the hazard rates, my question is: When can the condition that $P_B$ scales with N like $P_{fac}$ be met? Is this always the case for extensive systems? A clarification in the manuscript might be beneficial and could provide more clarity for the QMC community when to use the method.

Reply 4: We thank the referee for their insightful question. The complexity of our method scales as $C \sim O(\log(P_B)/\log(P_{fac}))$, where $P_B$ is the bound consensus probability, and $P_{fac}$ is the acceptance probability of the factorized Metropolis filter. This scaling depends on the energy profile of the updates. For a local update between $\tau_1$ and $\tau_2$ can be expressed as:

$$C\sim \frac{\sum_i\max|\Delta E_i(\tau_1-\tau_2)|}{\sum_i|\Delta E_i(\tau_1-\tau_2)|}$$
Therefore, the average complexity should be of $O(1)$, if the denominator and the nominator share the same scaling against $N$. For extensive system, both $\sum_i\max|\Delta E_i(\tau_1-\tau_2)|)$ and $\sum_i|\Delta E_i(\tau_1-\tau_2)|$ converges to finite values and a $O(1)$ complexity can be achieved. For sub-extensive, we expect the complexity to scale as $O(N^\alpha)$ with $0<\alpha<1$, similar to classical cases where interactions decay more slowly with distance. In the revised manuscript, we have clarified these scaling behaviors and the conditions under which $P_B$ and $P_{fac}$ scale similarly. This addition aims to provide better guidance for the QMC community on the applicability of our method.

Comment 5: While it was clearly introduced, the term "complexity" used in the manuscript could still be a bit misleading as it only refers to the complexity per update step (of which $N$ are performed). Changing it to something like "complexity per step" (also in the figures) could thus be more appropriate.

Reply 5: We sincerely thank the referee for this insightful suggestion. In the revised manuscript, we introduce a precise mathematical definition of Complexity to avoid ambiguity. For local update schemes described in this manuscript, we define Complexity per update, denoted as $C$, as the number of pairwise energy evaluations per local update:

$$C= # \text{pairwise energy evaluations per local update}.$$
This metric provides a direct and quantitative measure of the computational effort required by the algorithm, independent of the hardware setup. Since pairwise energy evaluations are typically the most computationally intensive step in simulating long-range interacting systems, this definition offers a meaningful proxy for the runtime per Monte Carlo sweep. This can be verified in the performance benchmark presented in the paper. For brevity, we refer to it as Complexity in the rest of the paper. We clarify these in the revised manuscript.

Comment 6: The authors mentioned that the clock factorized method does not affect the general dynamics of the worm algorithm. Does the clock factorized approach affect the measurement of Green's functions? It might be beneficial to (shortly) comment on this in the manuscript.

Reply 6: We appreciate the referee for this insightful question. The clock factorized method does not influence the measurement of Green's function in the worm algorithm. In this approach, Green's function can be measured by recording the histogram of space-time separation between the head and tail, and normalizing it by the relative ratio of the partition functions $Z_G$ and $Z$. Since the clock factorized QMC method does not change the configuration weights in either G-space or Z-space, the Green's function measurement remains unaffected.

We have included a brief comment on this in the revised manuscript to clarify this point.

Comment 7: I have to admit being a bit confused regarding the statements for frustrated systems in the discussion. Can the box technique here help reduce the QMC sign problem?

Reply 7: We thank the referee for raising this important point, and we would like to clarify the potential confusion. In the context of our discussion, frustrated systems specifically refer to systems with frustrated interactions in diagonal terms. The box technique can, to some extent, help manage these frustrated interactions in diagonal terms, like its application in classical systems, as demonstrated by Michel et al. (2019).

However, we note that the box technique does not address the QMC sign problem, which arises from off-diagonal terms. To avoid further confusion, we have clarified this distinction in the revised manuscript.

Reply to Minor Points:

We thank the referee for pointing out the typos and missing details. These issues have been carefully addressed and corrected in the revised manuscript.

Reply to Referee 2 We sincerely thank the referee for the thorough review and valuable suggestions. We follow the referee's suggestion and resubmit this manuscript to SciPost Phys Core. Below, we address the referee's specific comments.

Comment 1: "In chapter 4, for all examples, the authors present numerical results of the new algorithm for a quantity which is called "Complexity". I think it would be important to define this quantity precisely in an equation and refer to this equation and quantity whenever possible."

Reply 1: We sincerely thank the referee for this insightful suggestion. In the revised manuscript, we introduce a precise mathematical definition of Complexity to avoid ambiguity. For local update schemes described in this manuscript, we define Complexity per update, denoted as $C$, as the number of pairwise energy evaluations per local update:

$$C= # \text{pairwise energy evaluations per local update}.$$
This metric provides a direct and quantitative measure of the computational effort required by the algorithm, independent of the hardware setup. Since pairwise energy evaluations are typically the most computationally intensive step in simulating long-range interacting systems, this definition offers a meaningful proxy for the runtime per Monte Carlo sweep. This can be verified in the performance benchmark presented in the paper. For brevity, we refer to it as Complexity in the rest of the paper.

For conventional Metropolis-based algorithms, this quantity is of $O(N)$, whereas for our proposed algorithm, it approaches $O(1)$ in most cases. By explicitly referencing this definition throughout the revised manuscript, readers will be able to more easily understand, compare, and assess the computational efficiency of our algorithm.

Comment 2: "In chapter 4, for all examples, it would be important to also include physical quantities like the ground-state energy or order parameter and compare the obtained results quantitatively with the known ones from literature. Current simulations are done at rather large temperatures. How do the simulations perform by a proper scaling of temperature and length scales of systems when extracting quantum critical properties?"

Reply 2: We sincerely thank the referee for raising these important points. While we recognize the value of benchmarking physical quantities such as the ground-state energy or order parameters against known results, the primary focus of this manuscript is to introduce and validate the Clock Factorized PI-QMC methodology rather than to perform an exhaustive analysis of specific physical systems. Moreover, our algorithms can be further optimized in both update strategies and implementations for specific physical systems and parameters. As such, conducting a detailed benchmark of physical observables at this stage may not provide results representative of the full potential of our method and could not highlight the main purpose of this paper. A systematic analysis of quantum critical properties and physical observables for systems such as Rydberg atom arrays and long-range XXZ models is undergoing, and these quantities will be studied in depth.

To study quantum critical behavior in a path-integral QMC framework, one typically performs simulations at an inverse temperature that scales as the system size, for instance, $\beta \approx L^z$, where $z$ is the dynamical critical exponent of the quantum phase transition. The computational effort for such simulations generally scales linearly with $\beta$ due to the strictly short-range nature of interactions in the imaginary time dimension. The clock technique, specifically designed to address the complexity of long-range interactions in the spatial dimension, dramatically improves overall scaling performance.

For conventional Metropolis-based methods, the computational cost per sweep is $O(N^2)$ due to the $O(N^2)$ pairwise interactions in long-range systems. By contrast, the clock QMC method reduces this cost to $O(N\times C)$ per sweep. Thus, the overall scaling of the computational effort per sweep for a long-range interacting system of size $N$ at inverse temperature $\beta$ is: - $O(\beta N^2)$  for conventional algorithms - $O(\beta N \times C)$for the clock QMC method.

In this manuscript, we focus on simulations at fixed temperatures to emphasize the efficiency gains provided by the clock technique in handling long-range interactions. We have now included clarifications in the revised manuscript regarding the scaling of temperature and system size. This additional context should help readers better understand the rationale behind our approach.

Comment 3: "I am surprised that the authors do not refer to stochastic series expansion quantum Monte Carlo pioneered in Sandvik, A.W. Stochastic series expansion method for quantum Ising models with arbitrary interactions. Phys. Rev. E 2003, 68, 056701, which is heavily used for quantum systems with long-range interactions. For more bibliography please also check the recent review Entropy 2024, 26(5), 401 on Monte-Carlo approaches to long-range interactions in quantum systems, which in particular covers also the physics of the three examples discussed by the authors."

Reply 3: We thank the referee for pointing out the omission of these key references and for suggesting an additional bibliography. In the revised manuscript, we have included the recommended references: Sandvik's 2003 paper on the Stochastic Series Expansion (SSE) method (Phys. Rev. E 68, 056701) and the recent review on Monte Carlo approaches to long-range interactions in quantum systems published in Entropy 2024, 26(5), 401. We have included the suggested references in the revised manuscript.

Reply to Minor Points: We thank the referee for pointing out these typos and missing references. In the revised manuscript, we have addressed all the mentioned issues, including adding missing references and correcting typographical errors for better readability.

---

## Round 2 · List of Changes

1. At the beginning of Section 2, we add a short paragraph to outline Section 2 and highlight the key improvement of our method.
  2. In Section 2.3, we comment on the scaling of complexity and clarify that $O(1)$ complexity can be generally achieved for energy-extensive systems.
  3. In Section 4, we provide a more detailed definition of the Complexity of the update.
  4. In Section 4.2, we comment on the measurement of Green's function in the clock factorized worm algorithm.
  5. In Section 4.3, we comment on the acceptance ratio for the LRXXZ model and discuss possible solutions.
  6. In Section 5, we clarify the use of the box technique in a system with frustrated interactions in diagonal terms.
  7. In Section 5, we discuss our method's temperature scaling.
  8. Section 6 adds a paragraph acknowledging the SSE method in long-range interacting quantum systems.
  9. Fix typos and add missing references.

---

## Editorial Decision

published